# Locomotion modulates olfactory learning through proprioception in *C. elegans*

Xu Zhan ⓘ[1,5], Chao Chen ⓘ[2,3,5], Longgang Niu ⓘ[4], Xinran Du ⓘ[4], Ying Lei ⓘ[1], Rui Dan ⓘ[1], Zhao-Wen Wang ⓘ[4] ✉ & Ping Liu ⓘ[1] ✉

Locomotor activities can enhance learning, but the underlying circuit and synaptic mechanisms are largely unknown. Here we show that locomotion facilitates aversive olfactory learning in *C. elegans* by activating mechanoreceptors in motor neurons, and transmitting the proprioceptive information thus generated to locomotion interneurons through antidromic-rectifying gap junctions. The proprioceptive information serves to regulate experience-dependent activities and functional coupling of interneurons that process olfactory sensory information to produce the learning behavior. Genetic destruction of either the mechanoreceptors in motor neurons, the rectifying gap junctions between the motor neurons and locomotion interneurons, or specific inhibitory synapses among the interneurons impairs the aversive olfactory learning. We have thus uncovered an unexpected role of proprioception in a specific learning behavior as well as the circuit, synaptic, and gene bases for this function.

Locomotion has profound effects on brain functions, including learning and memory, by modulating brain-wide neuronal activity[1,2]. In humans, locomotor activity positively correlates with children's learning ability[3], and locomotion can instantly improve learning and memory[4,5]. In mice, locomotion enhances associative learning as well as neuronal activities in the cerebral cortex, hippocampus, and cerebellum, in a speed-dependent manner[1,6–8]. Furthermore, locomotion can reduce the risk of neurodegenerative diseases in elderly individuals, and substantially improve cognitive functions and reverse cognitive decline in both animal models and human patients with neurodegenerative diseases, indicating a conserved beneficial role of locomotion in learning and memory[9–11]. However, there are no known neural mechanisms mediating a coupling between locomotion and learning.

Locomotion may improve brain function through both acute and chronic effects[6,7,12–14]. Circulating myokines play a major role in mediating the slow pro-cognitive effect of locomotion[8,15]. However, the neural substrates mediating the acute pro-cognitive effect of locomotion are much less clear. Proprioceptive input is a good candidate because it acutely senses body movement and muscle stretch, and be transmitted to the central nervous system[16]. In mammals, proprioceptive signals may reach the cerebellum and cerebrum via the spinocerebellar tracts and the dorsal column-medial lemniscus pathway, respectively; while in insects, a fraction of proprioceptive axons projects to the central nervous system for proprioceptive feedback[16].

Learning is a universal function of the nervous system[17,18]. With only 302 neurons, *C. elegans* exhibits a range of learning behaviors[17–21]. For example, adult animals naturally prefer the smell of the pathogenic *Pseudomonas aeruginosa* strain PA14 but learn to avoid it after ingesting the bacterium for only a few hours[17,20,21]. Previous studies have mapped the primary neurons required for the PA14 aversive olfactory learning in the sensorimotor circuit (Supplementary Fig. 1a)[17,21–24]. Briefly, food odors are detected mainly by two bilateral pairs of sensory neurons: AWB and AWC, which signal to several downstream bilateral pairs of interneurons. Among the downstream interneurons, AVA, AIB, and RIM regulate the aversive olfactory

[1]Department of Pathophysiology, School of Basic Medicine and the Collaborative Innovation Center for Brain Science, Key Laboratory of Ministry of Education of China and Hubei Province for Neurological Disorders, Tongji Medical College, Huazhong University of Science and Technology, 430030 Wuhan, Hubei, China. [2]Department of Orthopaedics, Union Hospital, Tongji Medical College, Huazhong University of Science and Technology, 430022 Wuhan, Hubei, China. [3]Department of Orthopaedics, Hefeng Central Hospital, 445800 Enshi, Hubei, China. [4]Department of Neuroscience, University of Connecticut School of Medicine, Farmington, CT 06030, USA. [5]These authors contributed equally: Xu Zhan, Chao Chen. ✉e-mail: zwwang@uchc.edu; pingl@hust.edu.cn

learning of PA14, and exhibit synchronous activities due to interconnections through chemical synapses and gap junctions. Pre-exposure to PA14 suppresses and desynchronizes subsequent PA14-evoked sensory responses of AVA, AIB, and RIM to release a learned aversive behavior[22]. Five pairs of locomotion interneurons, including AVA, AVD, AVE, AVB, and PVC direct backward and forward movements, which are essential to aversive and attractive behaviors, respectively, with AVA and AVB being particularly important[25,26]. AVA initiate backward locomotion by controlling A-type cholinergic motor neurons (A-MNs) through both chemical synapses and gap junctions[27], whereas AVB promote forward locomotion only through gap junctions with B-type cholinergic motor neurons (B-MNs)[19]. Additionally, AVA and AVB may inhibit each other reciprocally[28,29].

The *C. elegans* motor circuit lacks specialized proprioceptive sensory neurons. L. Byerly and R.L. Russell speculated many years ago that A- and B-MNs might function as stretch sensors, utilizing their long asynaptic neurites[30]. Consistently, it has been shown that proprioception by B-MNs plays a crucial role in propagating the body bending wave from anterior to posterior regions, although molecular identities of the potential mechanoreceptors are unknown[31]. We recently found that GABAergic motor neurons (D-MNs), which receive chemical synaptic inputs from A- and B-MNs innervating contralateral muscles, are mechanosensitive and use UNC-8, a member of the DEG/ENaC/ASIC superfamily, for mechanosensation[32]. Thus, mechanosensation might be a common function of *C. elegans* motor neurons.

Here, we show that locomotion facilitates olfactory learning in a speed-dependent manner through proprioception. By taking advantage of the well-defined olfactory learning circuit for PA14 and using electrophysiological, genetic, $Ca^{2+}$ imaging, and optogenetic and chemogenetic techniques, we found that the interneurons AVA, AIB, and RIM enhance the olfactory learning by integrating proprioceptive information from motor neurons and exteroceptive (sensory) information from olfactory sensory neurons, and that AVB participate in the olfactory learning by interacting with AVA and RIM. In addition, we found that A- and B-MNs are both mechanosensitive; that gap junctions between AVB and B-MNs, like those between AVA and A-MNs reported in our previous study[27], are antidromic rectifiers; and that A- and B-MNs transmit locomotor information through the antidromic-rectifying gap junctions to the upstream interneurons AVA and AVB to tune their activities. Furthermore, we identified two DEG/ENaC/ASIC proteins (ASIC-1 and DEL-1) as key mechanoreceptors in A- and B-MNs, three innexins (UNC-7, UNC-9, and INX-19) as components of the gap junctions between AVB and B-MNs, a postsynaptic acetylcholine receptor (ACC-2) in AVA, and a postsynaptic tyramine receptor (LGC-55) in AVB. Thus, we have uncovered an unexpected role for proprioception in modulating learning and a neural circuit mediating the positive effect of locomotion on a specific form of learning.

## Results

### Locomotion enhances olfactory learning in a speed-dependent manner

To investigate whether and how the olfactory learning of PA14 may be affected by locomotion, we devised an experimental assay that allowed us to quantify the effect of locomotion speed on olfactory learning. In this assay, we expressed HisCl1, a histamine-gated chloride channel from *Drosophila*[33], specifically in body-wall muscle cells. A cohort of such transgenic animals, referred to as BWM-HisCl1 animals henceforth, were grown to the young adult stage in the presence of the standard food *E. coli* OP50, and then exposed to either OP50 or the pathogenic PA14 in the presence of histamine at various concentrations (one concentration per assay) for 6 h. Subsequently, some of the animals were used to quantify locomotion speed (speed test), while the rest were washed off and used for a choice test (Fig. 1a). The OP50- and PA14-exposed animals are referred to as naïve and trained animals, respectively.

In the speed test, we transferred animals to new nematode growth medium (NGM) plates without food and histamine (1 animal per plate) to quantify locomotion speed with an automated *C. elegans* tracking system[34]. The prior histamine application inhibited the locomotion speed of BWM-HisCl1 animals concentration-dependently (Supplementary Fig. 1b, c). In contrast, BWM-HisCl1 animals without prior histamine exposure and wild-type animals with prior histamine exposure did not show a decrease in locomotion speed (Supplementary Fig. 1b–d).

In the choice test, we placed animals in the center of a new histamine-free plate with OP50 and PA14 spotted on opposite sides, and quantified the results of the choice test by calculating a choice index and a learning index (Fig. 1a) as done by others[17,23]. Here, a positive choice index indicates an innate preference for PA14, whereas a positive learning index indicates a learned olfactory aversion to PA14. In BWM-HisCl1 animals with prior histamine treatment, the choice index was a positive value and invariable with respect to locomotion speed, whereas the learning index exhibited a positive correlation with locomotion speed (Fig. 1b). In contrast, the learning index was similar among wild type control, histamine-treated wild type, and BMW-HisCl1 animals without histamine treatment (Supplementary Fig. 1e). These results suggest that locomotion enhances the aversive olfactory learning in a speed-dependent manner but has no effect on either olfactory sensation or the naïve preference for PA14. Because histamine reduced locomotion speed of BWM-HisCl1 animals in a concentration-dependent manner with the half maximal inhibitory concentration ($IC_{50}$) around 5 mM (Supplementary Fig. 1b, c), and treating these animals with 5 mM histamine reduced locomotion speed to a level adequate to impair the learning index ($65.45 \pm 3.53 \ \mu m \ s^{-1}$) (Fig. 1b), we used this concentration in subsequent histamine experiments.

To confirm the effect of locomotion speed on olfactory learning, we tested whether inhibition of locomotion in wild type could impair olfactory learning. To this end, we added gelatin to NGM plates at various concentrations to impede animal locomotion[35]. Gelatin inhibited locomotion and olfactory learning concentration-dependently without altering the naïve choice index (Supplementary Fig. 2). Thus, olfactory learning is positively correlated with locomotion speed regardless of the means used to perturb locomotion.

### Locomotion modulates activities and functional coupling of AVA, AIB, and RIM

Exposures to both OP50 and PA14 induce a $Ca^{2+}$ drop in the interneurons AVA, AIB, and RIM. PA14 training produces the aversive olfactory learning behavior by attenuating the PA14-induced $Ca^{2+}$ drop[17,22-24]. To determine how locomotion regulates olfactory learning, we tested the effects of supernatants from PA14 and OP50 overnight cultures on GCaMP6 signals in AVA, AIB, and RIM. In these experiments, animals were imaged in solution with the head fixed to the bottom of the recording chamber by gluing, which allowed stable recordings of $Ca^{2+}$ signals in head neurons but free body bending. While the $Ca^{2+}$ drop induced by OP50 was indistinguishable between naïve and PA14-trained animals of both wild type and histamine-treated BWM-HisCl1 (Supplementary Fig. 3a, b), the $Ca^{2+}$ drop induced by PA14 was substantially smaller in PA14-trained than naïve animals in BWM-HisCl1 animals without prior histamine treatment (Fig. 1c), which is consistent with previous reports[22,23]. Notably, this difference was essentially abolished by histamine treatment (Fig. 1d). Quantitation demonstrated that histamine significantly reduced the PA14 training-induced changes in both amplitude and duration (Fig. 1e, f). Further analyses revealed that histamine minimized the difference in the $Ca^{2+}$ drop between naïve and trained animals by mainly acting on naïve animals (Supplementary Fig. 4a). These results suggest that locomotion may enhance olfactory learning by augmenting training-induced sensory responses of AVA, AIB, and RIM.

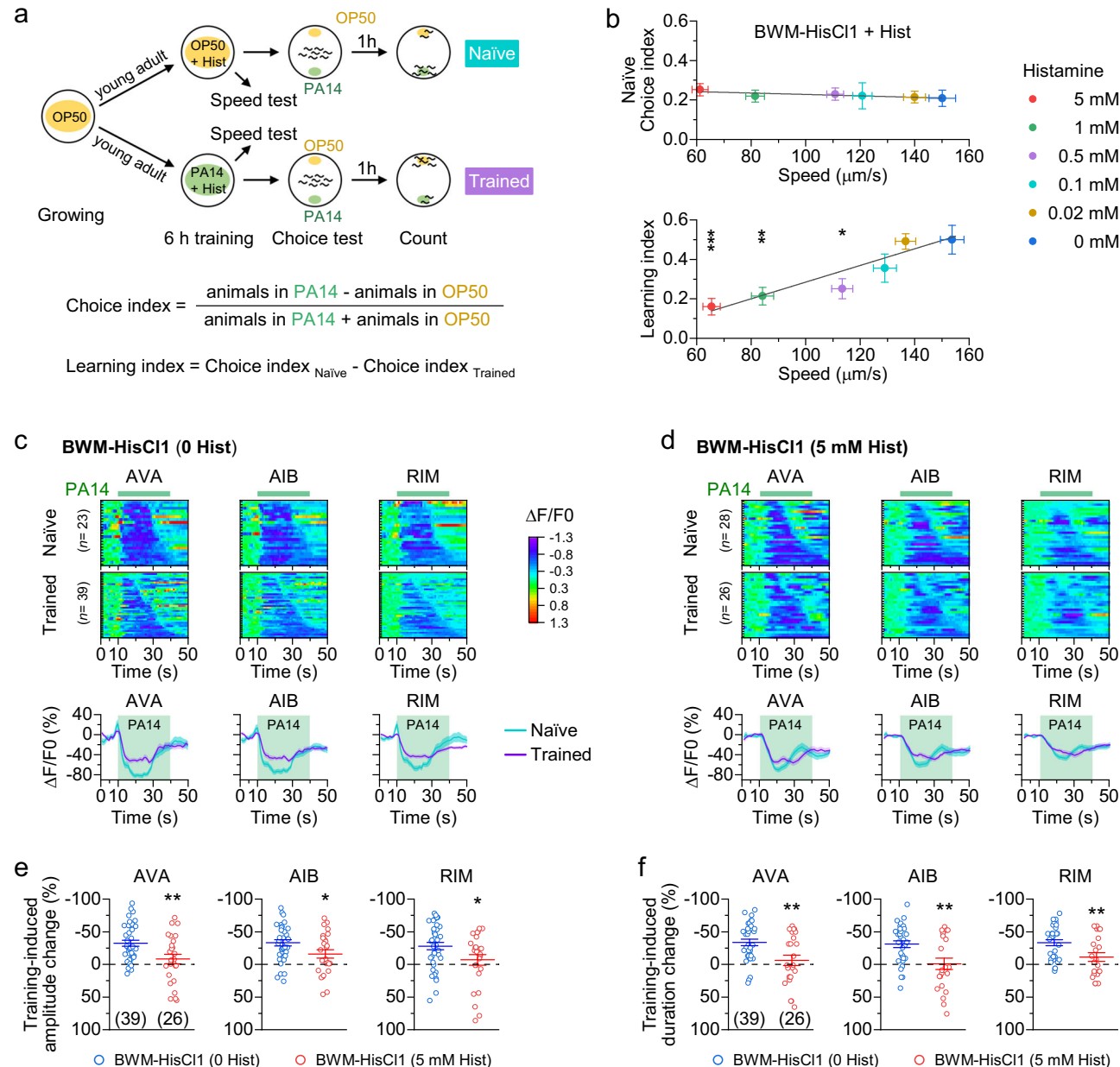

**Fig. 1 | Locomotion enhances olfactory learning speed-dependently. a** Diagram of the experimental procedures, and equations for calculating the choice index and learning index. **b** The learning index but not the naïve choice index depends on locomotion speed. A transgenic strain expressing HisCl1 in body-wall muscle cells (BWM-HisCl1, strain ID: PLX307) was used. Each data point represents results from 27 to 38 assays at one concentration of histamine. In each assay, 5 and 100–150 animals were used for the speed test and choice test, respectively (see Methods). Solid lines are linear fits to the data. *, **, and *** indicate $p < 0.05$, $p < 0.01$, and $p < 0.001$, respectively, compared with the 0-histamine control (one-way ANOVA with Tukey's post hoc test). Top, $p = 0.9758$, 1, 1, 1, and 1. Bottom, $p = 0.0006$, 0.0053, 0.013, 0.4685, and 1. **c, d** PA14-evoked Ca²⁺ drop in AVA, AIB, and RIM of naïve and trained BWM-HisCl1 animals (strain ID: PLX228) without and with prior histamine treatment. Top: Heatmaps of Ca²⁺ signal arranged by the response duration. Each row represents an individual animal's GCaMP6 signal over time. Bottom: The mean (solid line) and SEM (shaded area) of each group. **e, f** Comparison of training-induced amplitude and duration changes of the Ca²⁺ drop in AVA, AVB, and RIM between BWM-HisCl1 animals without and with prior histamine treatment. * and ** indicate $p < 0.05$ and $p < 0.01$, respectively (two-sided unpaired $t$ test). From left to right, $p = 0.004$, 0.0308, and 0.0296 (**e**), 0.0033, 0.0031, and 0.0074 (**f**). Brackets contain the number of animals tested (*n*). Data are shown as means ± SEM. Source data are provided as a Source Data file.

To confirm the effect of locomotion on the activities of the interneurons, we examined the effect of inhibiting muscle tone by histamine on spontaneous GCaMP6 signals in AVA, AIB, and RIM of BWM-HisCl1 animals. Indeed, histamine significantly reduced the frequency and amplitude of spontaneous intracellular Ca²⁺ transients (Fig. 2a). In addition, histamine disrupted synchrony of spontaneous Ca²⁺ transients among AVA, AIB, and RIM in BWM-HisCl1 but not wild-type animal (Fig. 2b, c). These results suggest that locomotion may enhance activities and synchrony of the interneurons.

## A- and B-MNs are mechanosensitive

*C. elegans* moves forward and backward by generating undulatory body waves through coordinated contraction and relaxation of dorsal and ventral body-wall muscles[25,26]. Thus, the locomotion-associated activities of AVA, AIB, and RIM might be due to muscle tone-related activation of mechanoreceptors. In mammals, muscle tension is sensed by Golgi tendon organs and muscle spindles, and the proprioceptive information is sent to the cerebral cortex through the dorsal column-medial lemniscus pathway and to the cerebellum

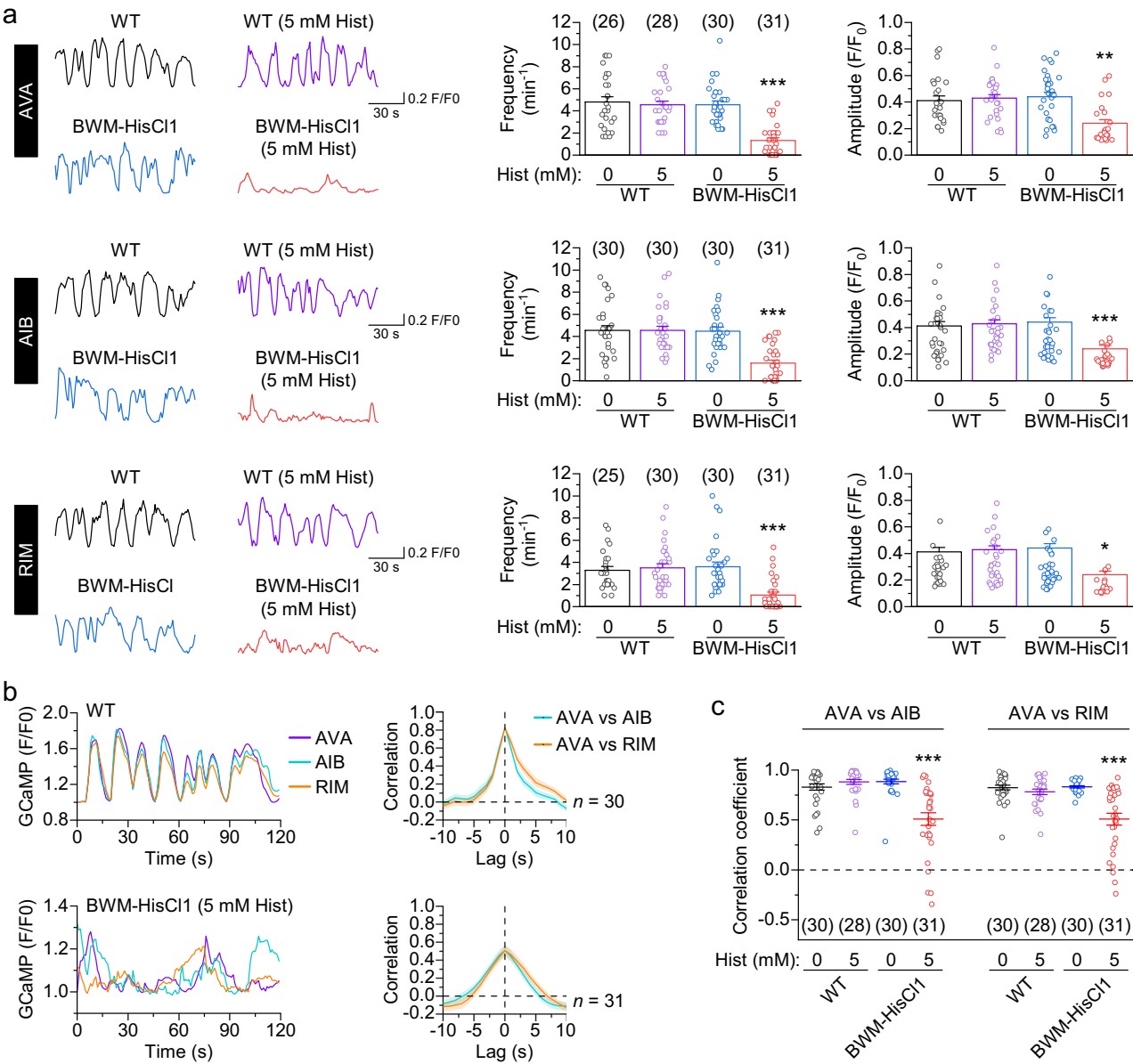

**Fig. 2 | Locomotion modulates activities and functional coupling of AVA, AIB, and RIM. a** Histamine inhibits spontaneous Ca²⁺ transients (measured by GCaMP6) in AVA, AIB, and RIM of animals expressing HisCl1 in body-wall muscles (BWM-HisCl1). Shown are sample traces (left) and comparisons of Ca²⁺ transient frequency and amplitude (right). WT = wild type. **b** Histamine impairs synchrony of spontaneous Ca²⁺ transients among AVA, AIB, and RIM in BWM-HisCl1 animals with prior histamine treatment. Shown are sample traces (left) and pairwise cross-correlation analyses (right) of GCaMP6 signals in WT and BWM-HisCl1 (5 mM Hist). **c** Comparison of GCaMP6 signal correlation coefficients. *, **, and *** indicate

$p < 0.05$, $p < 0.01$, and $p < 0.001$, respectively, compared with WT (0 Hist) (one-way ANOVA with Tukey's post hoc test). For frequency in (**a**), $p = 0.9623$, 0.9685, and 0 (AVA), 1, 0.9992, and < 0.0001 (AIB), 0.974, 0.9259, and 0.0002 (RIM). For amplitude in (**a**), $p = 0.9801$, 0.9169, and 0.0028 (AVA), 0.9492, 0.9237, and 0.0001 (AIB), 0.1913, 0.9962, and 0.0246 (RIM). For AVA vs AIB in (**c**), $p = 0.8183$, 0.7795, and < 0.0001. For AVA vs RIM in (**c**), $p = 0.8509$, 0.999, and < 0.0001. Brackets contain the number of animals tested ($n$). Data are shown as means ± SEM. Source data are provided as a Source Data file.

through the dorsal spinocerebellar and cuneocerebellar tracks[16,36]. Because *C. elegans* does not have the corresponding muscle structures and sensory neuron pathways but its motor neurons have been implicated in proprioception[30,31], we hypothesized that motor neurons themselves may detect and transmit proprioceptive signals to central neurons.

*C. elegans* A- and B-MNs are further divided into ventral (VA and VB) and dorsal (DA and DB) classes based on whether they innervate the ventral or dorsal muscles. To test our hypothesis, we chose the neighboring VA5, VB6, DA4, and DB4 as representatives of each class and tested the effect of mechanostimulation on motor neuron whole-cell currents. The mechanostimulation was produced by pressure

ejection (10 psi, 300 ms) of the bath solution through a glass pipette (~2 μm in tip diameter) aimed at specific regions of a recorded motor neuron[32]. We selected an injection pressure of 10 psi (equivalent to ~220 nN m⁻²) as it was the minimum pressure causing the maximal mechanoreceptor currents (MRCs) (Supplementary Fig. 5a), and falls within the range of estimated internal bending forces experienced by *C. elegans* during locomotion[37]. We chose an injection duration of 300 ms as it falls within the range of *C. elegans* bending duration (250 to 2,500 ms based on reported bending frequencies of approximately 0.4 to 4 events per second)[34,38]. This duration allowed us to capture both the peak and steady-state MRCs. In these experiments, the stimulation pipette was positioned perpendicular to the ventral nerve cord to

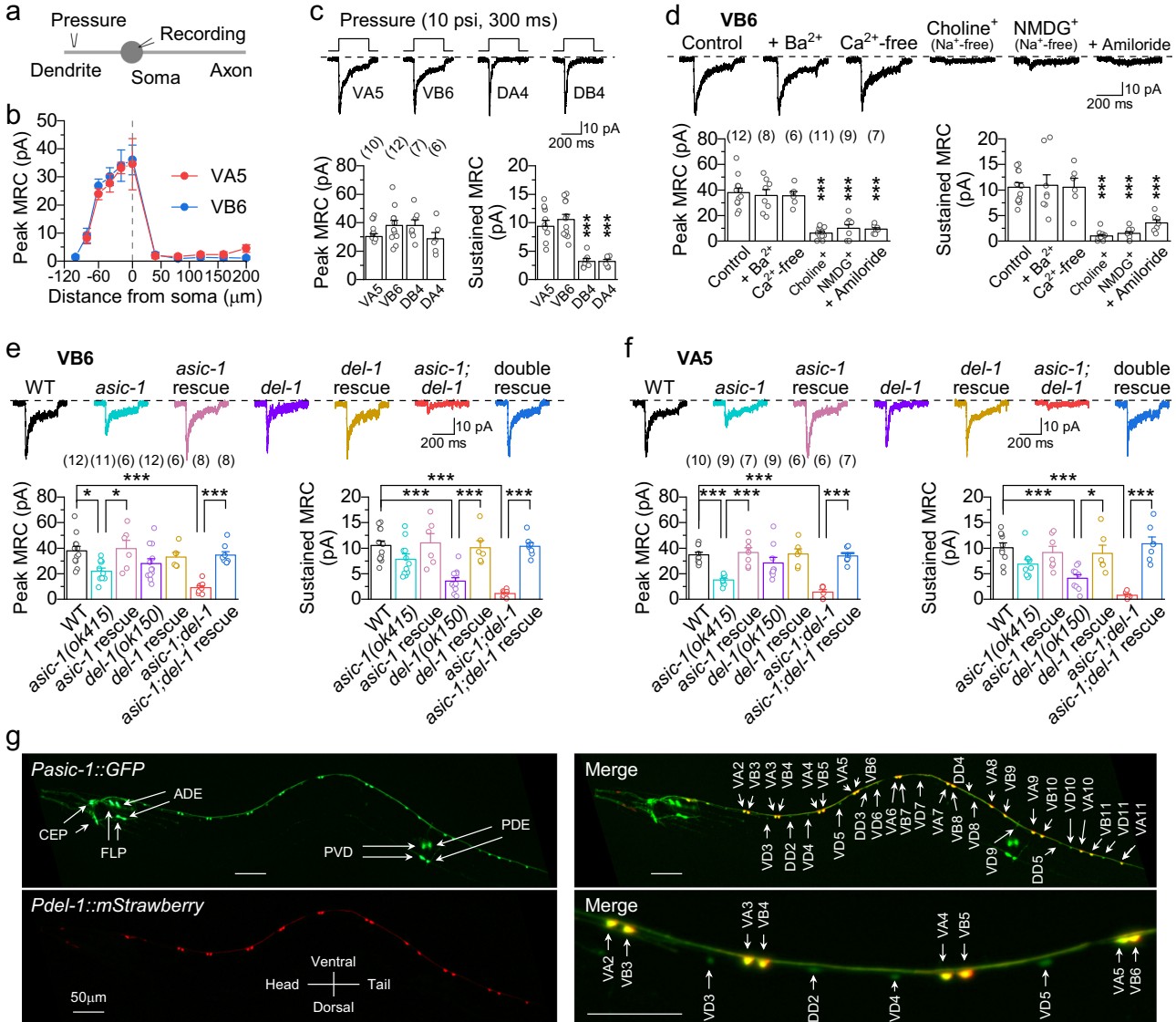

**Fig. 3 | ASIC-1 and DEL-1 form mechanoreceptors in VA and VB. a** Diagram showing how a stimulation pipette was positioned so that stimuli could be delivered to specific regions of a neuron. **b** Relationship between the mechanoreceptor currents (MRCs) and stimulation pipette location in VA5 and VB6. $n = 8$ for all data points of VA5 and 10-11 for data points of VB6. **c** MRCs in VA and VB differ from those in DA and DB stimulated at the soma. Shown are sample traces (top) and comparisons of peak MRC and sustained MRC (bottom). **d** MRCs in VB6 are eliminated by either application of amiloride or substitution of extracellular $Na^+$ by choline[+] or $NMDG^+$. **e, f** Mutations of *asic-1* and *del-1* impair MRCs in VB6 and VA5. Wild-type *asic-1* and *del-1* were expressed in ventral nerve cord cholinergic motor neurons using *Punc-17Δ1* in the rescue experiments. WT = wild type. **g** Expression of *del-1* is detected in VA and VB, and that of *asic-1* in additional neurons, including D-MNs, CEP, ADE, PDE, FLP, and PVD, in animals co-expressing *Pdel-1::mStrawberry* and *Pasic-1::GFP* transcriptional fusions. A similar pattern is observed in 23 animals.

Scale bar = 50 μm. *, and *** indicate $p < 0.05$ and $p < 0.001$, respectively (one-way ANOVA with Tukey's post hoc test). For peak MRC, $p = 0.8778, 0.9119$, and $0.6783$ (compared with VA5, **c**), $0.9943, 0.9946, 0, < 0.0001$, and $< 0.0001$ (compared with control, **d**), $0.0117, 0.9999, 0.2598, 0.9663, < 0.0001$, and $0.9911$ (compared with WT, **e**), $0.0293$ (*asic-1* vs *asic-1* rescue, **e**), $0.0003$ (*asic-1;del-1* vs double rescue, **e**), $0.0001, 0.9995, 0.6708, 1, < 0.0001$, and $1$ (compared with WT, **f**), $0.0002$ (*asic-1* vs *asic-1* rescue, **f**), $< 0.0001$ (*asic-1;del-1* vs double rescue, **f**). For sustained MRC, $p = 0.9789, < 0.0001$, and $< 0.0001$ (compared with VA5, **c**), $1, 1, < 0.0001, < 0.0001$, and $0.0005$ (compared with control, **d**), $0.2841, 0.9998, < 0.0001, 0.9999, < 0.0001$, and $1$ (compared with WT, **e**), $0.0005$ (*del-1* vs *del-1* rescue, **e**), $< 0.0001$ (*asic-1;del-1* vs double rescue, **e**), $0.2127, 0.9929, 0.0008, 0.9872, < 0.0001$, and $0.9983$ (compared with WT, **f**), $0.0392$ (*del-1* vs *del-1* rescue, **f**), $< 0.0001$ (*asic-1;del-1* vs double rescue, **f**). Brackets contain the number of cells recorded. Data are shown as means ± SEM. Source data are provided as a Source Data file.

target the soma, axon, or dendrite of a specific motor neuron, as shown in Fig. 3a.

For VA5 and VB6, which project axons posteriorly and anteriorly, respectively, and dendrites in the opposite directions[39], the stimulation electrode was aimed sequentially at multiple locations, including the soma, and the ventral nerve cord anterior and posterior to the soma at various distances (Fig. 3a). For DA4 and DB4, we only stimulated the soma and dendrite because their axons, which project circumferentially to the dorsal nerve cord (Supplementary Fig. 6a)[39], were disrupted in our preparations. In all the neurons examined, MRCs were observed upon stimulating the soma and the dendrite but not the axon, with the current amplitude being largest at the soma, and decreasing gradually in the dendrite with increasing distance from the soma (Fig. 3b and Supplementary Fig. 6b, c). We therefore only stimulated the soma in subsequent experiments. The diminution and eventual disappearance of MRCs at distal dendrites could be due to either fewer mechanoreceptors or stimulation beyond the dendritic terminal. The evoked MRCs of both VA5 and VB6 had an initial large peak followed by a sustained current (~1/3 of the peak current), while those of DA4 and DB4 had a similar peak current but a much smaller

sustained current (~1/10 of the peak current) compared with VA5 and VB6 (Fig. 3c). Nevertheless, both A- and B-MNs are mechanosensitive. Due to the proximity of neighboring motor neuron somata and the fact that motor neuron axons and dendrites run together in the ventral nerve cord, it was impossible to restrict the mechanical stimulation to a single motor neuron. However, the fact that MRCs could be induced only by stimulating specific regions of the motor neuron indicates that the recorded MRCs resulted from mechanoreceptors of the same neuron.

We further characterized the biophysical properties of MRCs using VB6 as the representative and observed several features. First, the MRC peak current was ejection pressure-dependent. It plateaued at 10-psi pressure, and displayed reduced latency and faster activation rate with increasing stimulation pressure (Supplementary Fig. 5a). Second, the MRC duration was linearly related to the stimulus duration, but the MRC peak current was not affected by the stimulus duration (Supplementary Fig. 5b). Third, MRCs showed desensitization to paired-pulse stimuli, with the desensitization gradually eliminated at increasing inter-pulse intervals (Supplementary Fig. 5c). Because these properties of MRCs are characteristic of mechanosensitive channels[40,41], the mechanosensitive responses are likely mediated by mechanosensitive channels.

## ASIC-1 and DEL-1 form mechanoreceptors in VA and VB

We next sought to identify the putative mechanosensitive channels in A- and B-MNs. In *C. elegans*, known mechanosensitive channels include members of the TRP and DEG/ENaC/ASIC families. The former are non-selective cation channels permeable to $Ca^{2+}$, whereas the latter are amiloride-sensitive $Na^+$ channels[42]. We found in VB6 that MRCs were unaffected by removing extracellular $Ca^{2+}$ or replacing it with $Ba^{2+}$ but essentially eliminated by applying amiloride or substituting extracellular $Na^+$ with an impermeant cation (choline$^+$ or NMDG$^+$) (Fig. 3d), suggesting that the putative mechanosensitive channels are most likely of the DEG/ENaC/ASIC type.

The *C. elegans* genome harbors 30 DEG/ENaC/ASIC genes[43]. By analyzing VB6 MRCs in mutants available for 25 such genes, we found that only *asic-1(ok415)* and *del-1(ok150)*, which are both deletion mutants, exhibited defective MRCs (Fig. 3e and Supplementary Fig. 7a). In both VB6 and VA5, the *asic-1* mutant showed a substantial decrease in the peak current but a modest reduction (statistically insignificant) in the sustained current, while the *del-1* mutant displayed a modest reduction (statistically insignificant) in the peak current but a substantial decrease in the sustained current compared with wild type (Fig. 3e, f). The deficient MRCs of the mutants were fully rescued by expressing corresponding wild-type genes in ventral cord cholinergic motor neurons under the control of a modified *unc-17*/vesicular acetylcholine transporter promoter (*Punc-17Δ1*, the letter *P* before a gene name represents promoter)[38], indicating that they resulted from *asic-1* and *del-1* mutations. In *asic-1(ok415);del-1(ok150)* double mutant, MRCs of VB6 and VA5 were essentially absent (Fig. 3e, f), but voltage-dependent currents of these two neurons were apparently normal (Supplementary Fig. 7b), suggesting that ASIC-1 and DEL-1 are the principal mechanosensitive channels in these neurons, and that mutations of *asic-1* and *del-1* do not have a global effect on ion channels.

To substantiate our findings from the electrophysiological analyses, we examined expression patterns of *asic-1* and *del-1* by co-expressing GFP and mStrawberry under the control of *Pasic-1* and *Pdel-1*, respectively. In transgenic animals, GFP and mStrawberry signals were detected in both VA and VB (Fig. 3g). While *del-1* expression was essentially restricted to VA and VB, similar to a previous report[44], *asic-1* expression was detected in additional neurons, including CEP, ADE, PDE, FLP, PVD, and all D-MNs (Fig. 3g). A previous study reported *asic-1* expression in only the first three of the additional neurons[45]. The detection of more *asic-1*-expressing neurons by this study is probably

because of the use of a longer *Pasic-1* (3.7 kb versus 2 kb). The detection of *asic-1* and *del-1* expression in all VA and VB supports their role as key mechanoreceptors in all these neurons.

Although ASIC-1 and DEL-1 are critical to the mechanosensation of VA5 and VB6, the mutations of *asic-1* and *del-1* had no effect on MRCs of DA4 and DB4 (Supplementary Fig. 6d, e), suggesting that the observed MRCs of DA and DB are mediated by other mechanosensitive channels. This conclusion is supported by the apparently different kinetics of evoked MRCs between V- and D-type cholinergic motor neurons (Fig. 3c).

## Mechanoreceptors are required for locomotion to modulate olfactory learning

The identification of *asic-1* and *del-1* as mechanoreceptors in VA and VB made it possible to determine the role of proprioception in olfactory learning. To this end, we performed three different experiments. Firstly, we examined whether a lack of both ASIC-1 and DEL-1 affects the activities of AVA, AIB, and RIM. We found that both the frequency and mean amplitude of spontaneous $Ca^{2+}$ transients were greatly reduced in the *asic-1;del-1* double mutant (Fig. 4a and Supplementary Fig. 8a, b). The temporal correlation of $Ca^{2+}$ transients among AVA, AIB, and RIM was also significantly compromised in the *asic-1;del-1* mutant compared with wild type (Fig. 4b). The $Ca^{2+}$ transient phenotypes of the double mutant could be rescued completely by co-expressing wild-type *asic-1* and *del-1* in VA and VB using *Pdel-1* (Fig. 4a, b and Supplementary Fig. 8a, b). Secondly, we tested the effect of *asic-1* and *del-1* mutations on the PA14 training-induced attenuation of the $Ca^{2+}$ drop in AVA, AIB, and RIM. The training-induced change was significantly smaller in amplitude and shorter in duration in the *asic-1;del-1* mutant compared with wild type, which could be rescued completely by co-expressing wild-type *asic-1* and *del-1* in VA and VB (Fig. 4c and Supplementary Fig. 8c, d). The primary effect of the *asic-1;del-1* double mutation was to reduce the $Ca^{2+}$ drop in naïve animals (Supplementary Fig. 4b). Similar to that in wild type (Supplementary Fig. 3a), PA14 training did not affect OP50-induced $Ca^{2+}$ drop in the *asic-1;del-1* mutant (Supplementary Fig. 3c). Finally, we assessed the olfactory learning ability of the *asic-1;del-1* mutant. The double mutant exhibited a normal naïve choice index but a significantly decreased learning index, which could be fully rescued by co-expressing wild-type *asic-1* and *del-1* in VA and VB (Fig. 4d). Moreover, the modulatory effect of locomotion speed on olfactory learning was greatly reduced in the double mutant compared with wild type (Fig. 4e). Collectively, these results suggest that locomotion regulates activities of the interneurons and enhances olfactory learning by stimulating mechanoreceptors in A- and B-MNs.

Results of a previous study suggest that proprioception mediated by putative mechanoreceptors in B-MNs is necessary for the propagation of body bending waves, which drive locomotion[31]. The identification of ASIC-1 and DEL-1 as mechanoreceptors in VA and VB allowed us to quantify the effect of proprioception on locomotion. We analyzed locomotor kinematics, including forward speed, backward speed, averaged speed, and the propensity of directional movements using our automated *C. elegans* tracking system[34]. Surprisingly, all these parameters were indistinguishable between wild type and either single or double mutant of *asic-1* and *del-1* (Supplementary Fig. 9). The results indicate that a defect of proprioception in VA and VB alone is insufficient to impair the apparent locomotor behaviors under our experimental conditions. However, we cannot exclude a role of motor neuron-mediated proprioception in locomotion based on these results because mechanoreceptors of DA, DB, and D-MNs remained intact in these animals.

As locomotion speed can influence olfactory learning by activating mechanoreceptors in motor neurons, we were curious about how higher speeds might lead to stronger proprioception. We hypothesized that faster locomotion could enhance

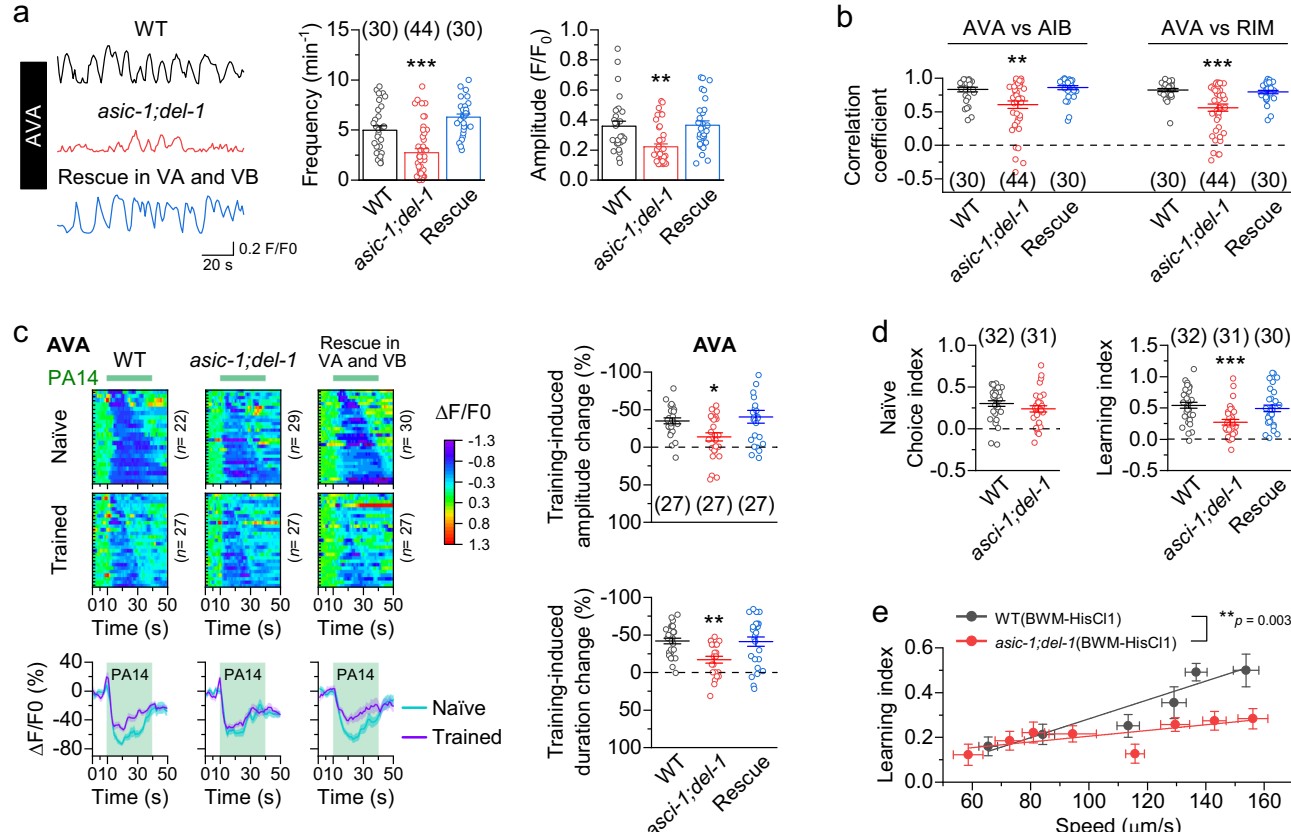

**Fig. 4 | ASIC-1 and DEL-1 mechanoreceptors are required for locomotion to modulate olfactory learning. a, b** *asic-1(ok415);del-1(ok150)* mutant shows great decreases in the frequency and amplitude of $Ca^{2+}$ transients in AVA, and in the correlation coefficients of $Ca^{2+}$ transients among AVA, AIB, and RIM compared with wild type (WT). The mutant phenotypes are rescued by expressing wild-type *asic-1* and *del-1* cDNAs in VA and VB using *Pdel-1*. **c** The *asic-1;del-1* mutant displays less PA14 training-induced change of the $Ca^{2+}$ drop in AVA than WT, and this phenotype is rescued in the strain expressing wild-type *asic-1* and *del-1* cDNAs in VA and VB. Shown are the heatmaps arranged by the response duration (top left), the mean (solid line) and SEM (shaded area) of each group, and statistical comparisons of the training-induced changes of the $Ca^{2+}$ drop (right). **d** The learning index but not the naïve choice index is significantly reduced in the *asic-1;del-1* mutant compared with WT, and the mutant phenotype is rescued in the strain expressing wild-type *asic-1* and *del-1* cDNAs in VA and VB. **e** The enhancing effect of locomotion speed on the learning index is greatly reduced in the *asic-1;del-1* mutant compared with WT. The transgenic animals expressing HisCl1 in body-wall muscle cells (BWM-HisCl1) were used (strain IDs: PLX310 and PLX307). Solid lines are linear fits to the data. WT data in Fig. 1b are redisplayed here. Each data point of the *asic-1;del-1* mutant group represents results from 28-32 assays using one concentration of histamine (from left to right are: 5 mM, 3 mM, 1 mM, 0.7 mM, 0.5 mM, 0.1 mM, 0.02 mM, and 0). *, **, and *** indicate $p < 0.05$, $p < 0.01$, and $p < 0.001$, respectively, based on either one-way ANOVA with Tukey's post hoc test (**a–d**), two-sided unpaired *t* test (**d**), or two-sided Fit comparison with *F*-test (**e**). Compared with WT, $p = 0.0004$ and 0.0798 (frequency, **a**), 0.0015 and 0.9873 (amplitude, **a**), 0.0022 and 0.9172 (AVA vs AIB, **b**), < 0.0001 and 0.8852 (AVA vs RIM, **b**), 0.041 and 0.8101 (top, **c**), 0.0025 and 0.9922 (bottom, **c**), 0.2054 (left, **d**), 0.0005 and 0.7682 (right, **d**). Brackets contain the number of animals tested (**a–c**) or independent assays (**d**). Data are shown as means ± SEM. Source data are provided as a Source Data file.

proprioception by increasing the bending amplitude and/or frequency of *C. elegans*. To test this, we reduced the locomotion speed of wild-type and BWM-HisCl1 animals to varying degrees by adding different concentrations of gelatin and histamine to assay plates, respectively. We then measured the body amplitude, a proxy for bending amplitude, and the bending frequency of the animals using our automated *C. elegans* tracker[34]. Our findings showed that the bending frequency increased with higher locomotion speed, while the body amplitude remained constant (Supplementary Fig. 10). These results suggest that locomotion may boost proprioception by elevating the frequency of mechanoreceptor activation rather than the magnitude, at least under our experimental conditions.

### Gap junctions are important to locomotion-dependent olfactory learning

To understand the neural circuit basis of locomotion-dependent olfactory learning, we explored the potential roles of gap junctions between locomotion interneurons and motor neurons. We previously found that the gap junctions between AVA and A-MNs only allow antidromic current from A-MNs to AVA, and are formed by the innexin UNC-7 in AVA and the innexin UNC-9 in A-MNs[27]. We thus began by characterizing the biophysical properties of the gap junctions between AVB and B-MNs. In response to bidirectional (positive and negative) and symmetric transjunctional voltage ($V_j$) steps in dual-whole cell voltage clamp experiments, junctional currents ($I_j$) flowed predominantly from VB6 into AVB (Fig. 5a, b), suggesting that the gap junctions strongly favor retrograde currents from B-MNs to the upstream AVB interneurons. This finding was confirmed by recording the $I_j$ between AVB and DB4, a B-MN innervating dorsal muscles (Supplementary Fig. 11a, b).

We next investigated molecular compositions of the gap junctions between AVB and B-MNs. Analyses of mutant behaviors and expression patterns of UNC-7 and UNC-9 have led to the suggestion that the gap junctions between AVB and B-MNs consist of UNC-7 in AVB and UNC-9 in B-MNs[46]. Consistently, we found that the $I_j$ between AVB and VB6 were much (by ~90%) smaller in the putative null mutants *unc-7(e5)* and *unc-9(fc16)*[46] than in wild type, and that this phenotype could be

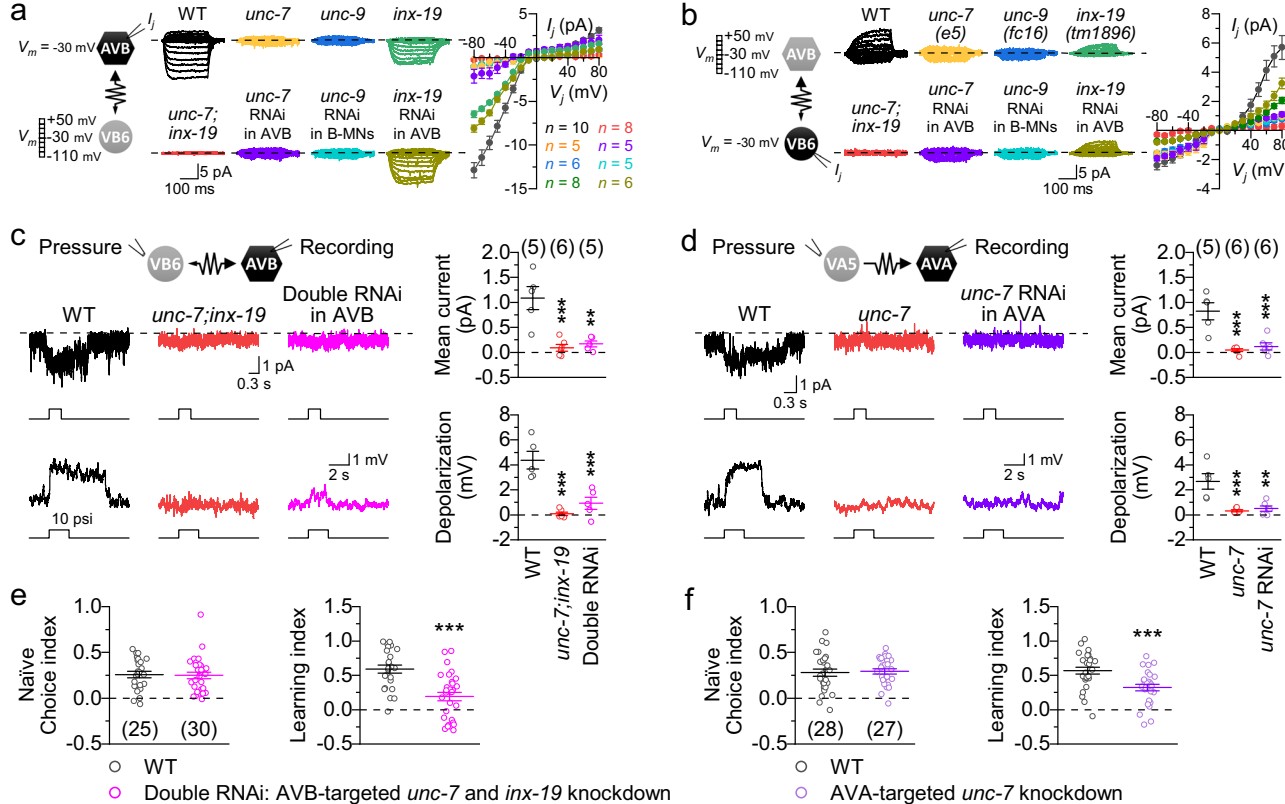

**Fig. 5 | Antidromic-rectifying gap junctions between locomotion interneurons and motor neurons facilitate olfactory learning. a, b** Gap junctions between AVB and VB6 conduct current predominantly in the antidromic direction, and require UNC-7 and INX-19 in AVB and UNC-9 in B-MNs to function. Junctional currents ($I_j$) were recorded from AVB or VB6. Left: Diagrams of the recording protocol. $V_j$ = transjunctional voltage (see Methods). Right: Sample $I_j$ traces and $I_j$–$V_j$ relationships. RNAi was done using *Psra-11* (AVB) or *Pacr-5* (B-MNs). WT = wild type. **c** Stretch stimulation of VB6 evokes an inward current in AVB and causes AVB depolarization in WT, but not in either the *unc-7;inx-19* mutant or the AVB-targeted *unc-7* and *inx-19* knockdown strain (Double RNAi). **d** Stretch stimulation of VA5 evokes an inward current in AVA and causes AVA depolarization in WT, but not in either the *unc-7* mutant or the AVA-targeted *unc-7* knockdown strain (*unc-7* RNAi). In (**c, d**), the low-Cl⁻ pipette solution (see Methods) was used, and the holding voltage was 0 mV for current recording. **e, f** Knockdown of *unc-7* and *inx-19* in AVB or *unc-7* in AVA significantly decreases the olfactory learning index but not the naïve choice index. ** and *** indicate $p < 0.05$ and $p < 0.001$, respectively, based on either one-way ANOVA with Tukey's post hoc test (**c, d**) or two-sided unpaired $t$ test (**e, f**). Compared with WT, $p = 0.0004$ and $0.0012$ (top, **c**), $< 0.0001$ and $0.0007$ (bottom, **c**), $0.0002$ and $0.0005$ (top, **d**), $0.0008$ and $0.0017$ (bottom, **d**), $0.8503$ (left, **e**), $< 0.0001$ (right, **e**), $0.7805$ (left, **f**), $0.0007$ (right, **f**). Brackets contain the number of cell pairs or cells recorded (**a–d**) or independent assays (**e, f**). Data are shown as mean ± SEM. Source data are provided as a Source Data file.

recapitulated in wild type by knockdown (RNAi) of either *unc-7* in AVB or *unc-9* in B-MNs (Fig. 5a, b).

The persistence of small and non-rectifying $I_j$ in both the *unc-7* and *unc-9* mutants prompted us to search for other innexins contributing to the electrical coupling in AVB and B-MNs. Because AVB also express INX-19 (also known as NSY-5)[46], we investigated whether this innexin plays a role. Indeed, we found that the $I_j$ between AVB and VB6 were significantly smaller in either *inx-19(tm1896)*, a loss-of-function mutant resulting from deletion[47], or wild-type animals with AVB-targeted *inx-19* RNAi (Fig. 5a, b), suggesting that INX-19 in AVB also contributes to the electrical coupling. To confirm the roles of UNC-7 and INX-19 in AVB, we recorded the $I_j$ between AVB and VB6, and between AVB and DB4 in the *unc-7(e5);inx-19(tm1896)* double mutant. We found that the electrical coupling was eradicated in the double mutant (Fig. 5a, b and Supplementary Fig. 11a, b), suggesting that UNC-7 and INX-19 play a crucial role in the coupling by acting in AVB.

We then tried to identify the innexin(s) that function with UNC-9 in B-MNs. Besides *unc-9*, previous studies have identified six other innexins expressed in B-MNs, including *inx-1, inx-3, inx-7, inx-10, inx-12,* and *inx-14*[48,49]. We assessed their potential roles in the electrical coupling by analyzing the $I_j$ between AVB and VB6 in animals deficient in the innexins. We used available mutants for *inx-1, inx-7, inx-10,* and *inx-14* but created cholinergic motor neuron-targeted knockdown strains for *inx-3* and *inx-12* because their mutants are non-viable. However,

none of the mutants or knockdown strains showed deficient $I_j$ (Supplementary Fig. 11c, d), suggesting that another innexin(s) that was not identified in the previous studies might contribute to the electrical coupling.

With the knowledge of biophysical properties and molecular compositions of the gap junctions (between AVA and A-MNs and between AVB and B-MNs) at hand, we tested whether stretch activation of cholinergic motor neurons may activate the locomotion interneurons, and whether the gap junctions between them may regulate olfactory learning. Stretch stimulation of VB6 and VA5 by pressure ejection (10 psi, 300 ms or 2 s) in wild type induced inward current and membrane depolarization in AVB and AVA, respectively (Fig. 5c, d). These effects of VB6 stimulation on AVB were not detected in *unc-7(e5);inx-19(tm1896)* and greatly decreased in a transgenic strain with AVB-targeted knockdown of both *unc-7* and *inx-19* (Fig. 5c); while those of VA5 stimulation on AVA were not detected in both *unc-7(e5)* and a transgenic strain with AVA-targeted *unc-7* knockdown (Fig. 5d). These results indicate that stretch stimulation of A- and B-MNs is sufficient to activate AVA and AVB through the antidromic-rectifying gap junctions. Notably, while the naïve choice index was unaffected, the learning index was significantly reduced in the knockdown strains (Fig. 5e, f), suggesting that the antidromic-rectifying gap junctions play a crucial role in relaying the proprioceptive information from cholinergic motor neurons to the locomotion interneurons.

To determine the combined effects of deficient gap junctions between AVA and A-MNs and between AVB and B-MNs on activities of the interneurons critical to olfactory learning, we created a triple RNAi strain with AVA-targeted knockdown of *unc-7* and AVB-targeted knockdown of both *unc-7* and *inx-19*, and analyzed the effect of PA14 training on Ca²⁺ transients in AVA, AIB, and RIM. The triple RNAi strain exhibited great reductions in both the magnitude and duration of the PA14 training-induced Ca²⁺ drop in AVA, AIB, and RIM when compared to wild type (Fig. 6a–d), predominantly resulting from a reduced Ca²⁺ drop in naïve animals (Supplementary Fig. 4c). Additionally, the triple RNAi strain showed approximately 50% decrease in the learning index

without a change in the naïve choice index (Fig. 6e), and a greatly reduced effect of locomotion speed on the learning index compared with wild type (Fig. 6f). These results lend further support to the notion that the gap junctions between the locomotion interneurons and cholinergic motor neurons may mediate the effect of locomotion on olfactory learning.

Both *unc-7* and *unc-9* are expressed in many neurons, while *unc-9* is additionally expressed in body-wall muscle cells[48,49]. Loss-of-function mutations in either *unc-7* or *unc-9* lead to locomotion defects[46]. Consequently, we questioned whether the observed olfactory learning defect in the various innexin knockdown strains was

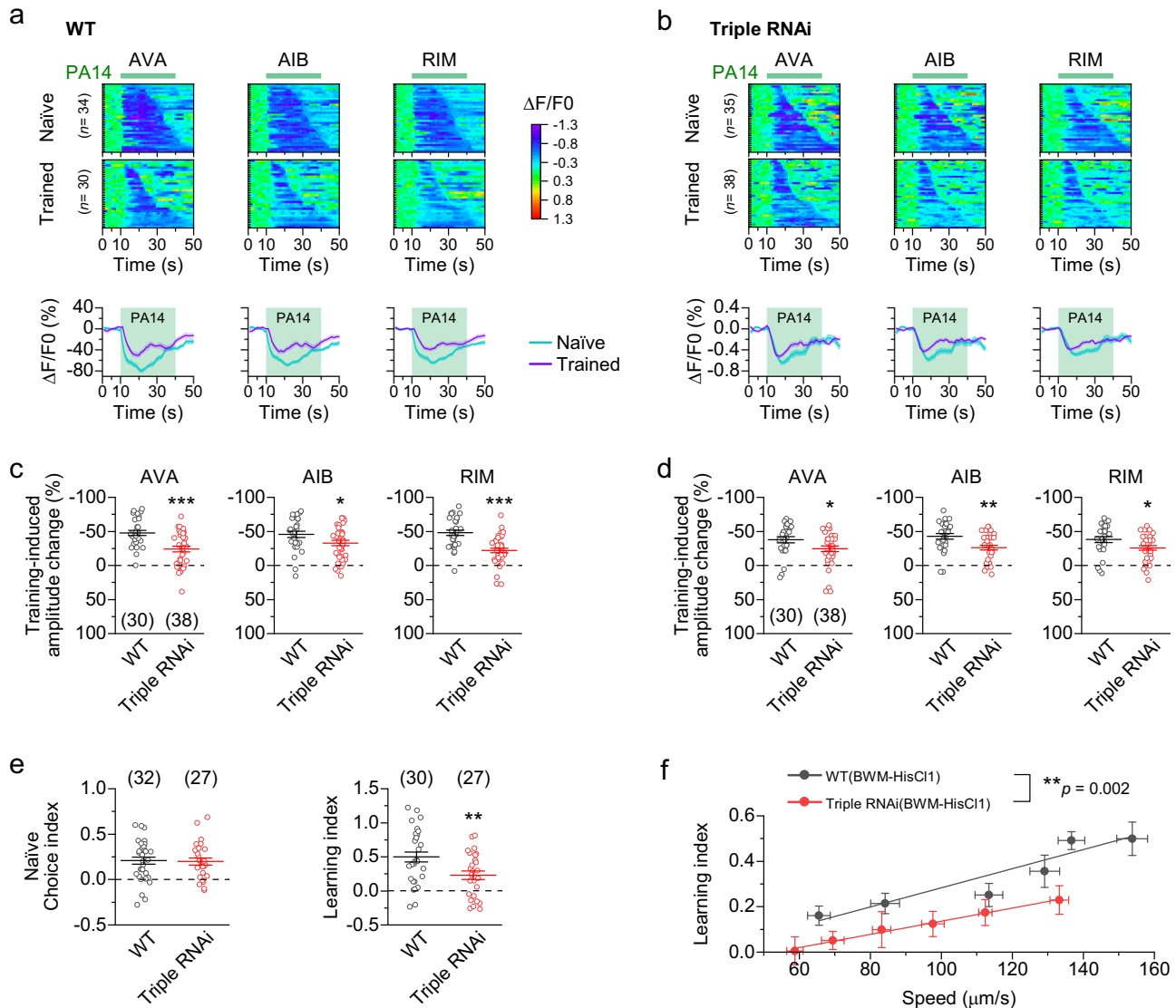

Triple RNAi: knockdown of *unc-7* and *inx-19* in AVB and *unc-7* in AVA

**Fig. 6 | Disruption of gap junctions between locomotion interneurons and motor neurons reduces PA14 training-induced Ca²⁺ changes in AVA, AIB, and RIM.** Results were obtained from a triple knockdown strain (*unc-7* and *inx-19* in AVB, and *unc-7* in AVA) expressing GCaMP6 in selected neurons. **a, b** PA14-induced Ca²⁺ drop in naïve and trained animals of wild type (WT) and the triple RNAi strain. Top: Heatmaps arranged by the response duration. Bottom: The mean (solid line) and SEM (shaded area) of each group. **c, d** Comparison of training-induced amplitude and duration changes of the Ca²⁺ drop between WT and the triple RNAi strain. **e** Comparison of the learning index and the naïve choice index between WT and the triple RNAi strain. **f** Relationships between locomotion speed and the learning index in WT and the triple RNAi animals expressing HisCl1 in body-wall

muscle cells (BWM-HisCl1, strain IDs: PLX307 and PLX454). Solid lines are linear fits to the data. WT data is a redisplay of those in Fig. 1b. Each data point of the triple RNAi group represents results from 25–28 assays using one concentration of histamine (from left to right are: 0.8 mM, 0.7 mM, 0.4 mM, 0.1 mM, 0.01 mM, and 0). *, **, and *** indicate $p < 0.05$, $p < 0.01$, and $p < 0.001$, respectively, compared with WT, based on either two-sided unpaired *t* test (**c**–**e**) or two-sided Fit comparison with *F*-test (**f**). From left to right, $p$ = < 0.0001, 0.0338, and < 0.0001 (**c**), 0.0343, 0.0029, and 0.0267 (**d**), 0.8618 and 0.0074 (**e**). Brackets contain the number of animals tested (**a**–**d**) or independent assays (**e, f**). Data are shown as means ± SEM. Source data are provided as a Source Data file.

partially attributed to reduced locomotion speed. To address this query, we compared the locomotion speed among different strains: wild type, AVB-targeted *unc-7* and *inx-19* knockdown strain, AVA-targeted *unc-7* knockdown strain, and the triple RNAi strain. The AVA-targeted *unc-7* knockdown strain exhibited a locomotion speed similar to wild type (142.63 μm s$^{-1}$ vs. 151.74 μm s$^{-1}$, $p = 0.529$). However, both the AVB-targeted *unc-7* and *inx-19* knockdown strain (135.61 μm s$^{-1}$) and the triple RNAi strain (133.24 μm s$^{-1}$) displayed significantly slower locomotion speed compared to wild type ($p = 0.017$ and 0.011, respectively). The magnitude of locomotion reduction in both knockdown strains was similar to that caused by 0.02 mM histamine in the BWM-HisCl1 strain, which had no significant effect on the olfactory learning index (Fig. 1b). Therefore, the olfactory learning defect observed in the knockdown strains resulted from the blockade of the antidromic junctional current rather than the inhibition of locomotion.

## AVA and AVB regulate olfactory learning by interacting with RIM

AVA regulate the aversive olfactory learning of PA14 by interacting with RIM through both excitatory chemical synapses and gap junctions[22,50]. Since AVB have chemical synapses with both AVA and RIM[30] (Supplementary Fig. 1a), and AVB activation by antidromic $I_j$ facilitated olfactory learning (Fig. 5c, e), we wondered whether AVB interact with AVA and RIM to regulate the olfactory learning. Consistently, acute inhibition of AVB by applying histamine (5 mM) to a transgenic strain expressing HisCl1 specifically in AVB greatly reduced both the training-induced attenuation of the Ca$^{2+}$ drop in AVA (Fig. 7a, b) and the learning index without affecting the naïve choice index (Fig. 7c). The inhibition of AVB primarily affected training-induced Ca$^{2+}$ changes by acting on naïve animals (Supplementary Fig. 4d). These results resemble the effects of locomotion inhibition (Fig. 1b–f) and mechanoreceptor mutations (Fig. 4c, d), suggesting that AVB contribute to olfactory learning by regulating the training-induced sensory responses in AVA.

To answer the question of how AVB might regulate AVA, we first analyzed spontaneous Ca$^{2+}$ transients in AVA and AVB of a transgenic strain expressing GCaMP6 in them. We found that Ca$^{2+}$ transients in AVA and AVB were strongly anticorrelated (Fig. 7d), in agreement with studies suggesting reciprocal inhibitory connections between them (6 from AVA to AVB and 33 from AVB to AVA)[29,51]. Next, we performed voltage-clamp experiments with two independent transgenic strains expressing channelrhodopsin-2 (ChR2) in AVA and AVB, respectively, to determine whether AVA and AVB might inhibit each other. We observed strong optogenetically evoked inhibitory currents in AVA but not in AVB (Fig. 7e), consistent with the existence of more chemical synapses from AVB to AVA than from AVA to AVB[51]. Thus, our results indicate that AVB inhibit AVA through chemical synapses.

As AVB use acetylcholine (ACh) as a neurotransmitter[52], we suspected that the postsynaptic receptor in AVA is an ACh-gated Cl$^-$ channel. The *C. elegans* genome harbors eight genes of putative ACh-gated Cl$^-$ channels, including *acc-1*, *acc-2*, *acc-3*, *acc-4*, *lgc-46*, *lgc-47*, *lgc-48*, and *lgc-49*[43]. However, our previous studies suggest that LGC-46 may form ACh-gated cation channels in motor neurons[27,32]. To identify the putative ACh-gated Cl$^-$ channel in AVA, we compared the effect of exogenous ACh (1 mM) on AVA whole-cell currents between wild type and mutants of the candidate genes. In this assay, ACh was applied by pressure-ejection (2 psi, 30 ms) through a glass pipette aimed at AVA. As expected, ACh elicited an outward current in AVA of wild type (Fig. 7f). The response of AVA to exogenous ACh was normal in mutants of all the candidate genes except *acc-2* (Fig. 7f and Supplementary Fig. 12). In *acc-2(ok2216)*, which is a deletion mutant, the ACh-induced outward current in AVA was reduced by more than 50%, and the optogenetically evoked AVA outward current observed in wild type was completely absent (Fig. 7f, g). These mutant phenotypes could be completely rescued by expressing wild-type *acc-2* specifically in AVA

and recapitulated in wild type by AVA-targeted *acc-2* knockdown (Fig. 7f, g). Collectively, these results indicate that ACC-2 is the postsynaptic ACh receptor mediating the chemical synaptic transmission from AVB to AVA. The persistence of some ACh-induced outward current in AVA of *acc-2(ok2216)* was probably due to the activation of another Cl$^-$ channel that is either extrasynaptic or postsynaptic to other cholinergic neurons.

We then created an AVA-targeted *acc-2* knockdown strain and tested whether the anticorrelation of activity between AVA and AVB and the olfactory learning behavior are deficient in this strain. Indeed, the *acc-2* knockdown impaired the anticorrelation between AVA and AVB (Fig. 7h) and reduced the olfactory learning index without affecting the naïve choice index (Fig. 7i). Thus, ACC-2-dependent inhibition of AVA by AVB is necessary for their anticorrelation and olfactory learning.

We next examined the functional relationship between AVB and RIM. The bilateral pair of RIM interneurons is the only two tyraminergic neurons in the *C. elegans* sensorimotor circuit[53]. Previous studies have predicted that RIM may release tyramine to inhibit AVB through LGC-55, a tyramine-gated Cl$^-$ channel, based on mutant behaviors and LGC-55 expression in AVB[53,54]. Consistently, we found that exogenous tyramine (200 μM) elicited an inhibitory current in AVB of wild type but not *lgc-55(n4331)*, a deletion mutant, and that the defective response of the mutant could be rescued by expressing wild-type *lgc-55* in AVB and recapitulated in wild type by AVB-targeted *lgc-55* knockdown (Fig. 7j). These results indicate that RIM inhibit AVB by activating the postsynaptic tyramine receptor LGC-55.

We then assessed the role of the RIM-AVB inhibitory synapse in olfactory learning by testing the effect of AVB-targeted *lgc-55* knockdown. The *lgc-55* knockdown strain showed a significantly reduced olfactory learning index without a change in the naïve choice index (Fig. 7k), indicating an indispensable role of the RIM-AVB inhibitory synapse in olfactory learning. This result reinforces our conclusion that AVB play an especially important role in olfactory learning. It is also in agreement with a previous finding that the tyraminergic output of RIM is required for the olfactory learning of PA14[17]. Notably, the defective learning caused by AVB-targeted *lgc-55* knockdown was accompanied by a reduced anticorrelation of activity between AVA and AVB (Fig. 7l). Together, these results demonstrate that AVB regulate olfactory learning by interacting with AVA and RIM through the RIM-AVB tyraminergic inhibitory synapse and the AVB-AVA cholinergic inhibitory synapse.

ASH are nociceptive sensory neurons that normally do not respond to the stimulation of PA14. However, PA14 training changes ASH from being nonresponsive to being strongly activated by PA14, and genetic ablation of ASH impairs olfactory learning[22]. Because ASH also provide synaptic inputs to AVA, AIB, RIM, and AVB[22,29,30], we examined whether the proprioception-mediated olfactory learning involves changes in ASH activity by Ca$^{2+}$ imaging. Consistent with the previous report[22], we observed little ASH responses to PA14 in naïve animals but strong activation by PA14 in trained animals (Supplementary Fig. 13a, b). However, neither locomotion inhibition by histamine in BWM-HisCl1 animals, nor elimination of proprioception by the *asic-1;del-1* double mutation or disruption of the antidromic $I_j$ from the motor neurons to the locomotion interneuron by the triple knockdown (*unc-7* in AVA, and *unc-7* and *inx-19* in AVB) had a detectable effect on the response of ASH to PA14 in both naïve and trained animals (Supplementary Fig. 13a–d). These results indicate that the effect of locomotion on olfactory learning does not involve changes in ASH activity.

## Proprioception also regulates olfactory adaptation

Our findings with PA14 raised the question of whether proprioception also plays a role in other olfactory learning behaviors. To answer this question, we chose to test the effects of *asic-1* and *del-1* mutations on

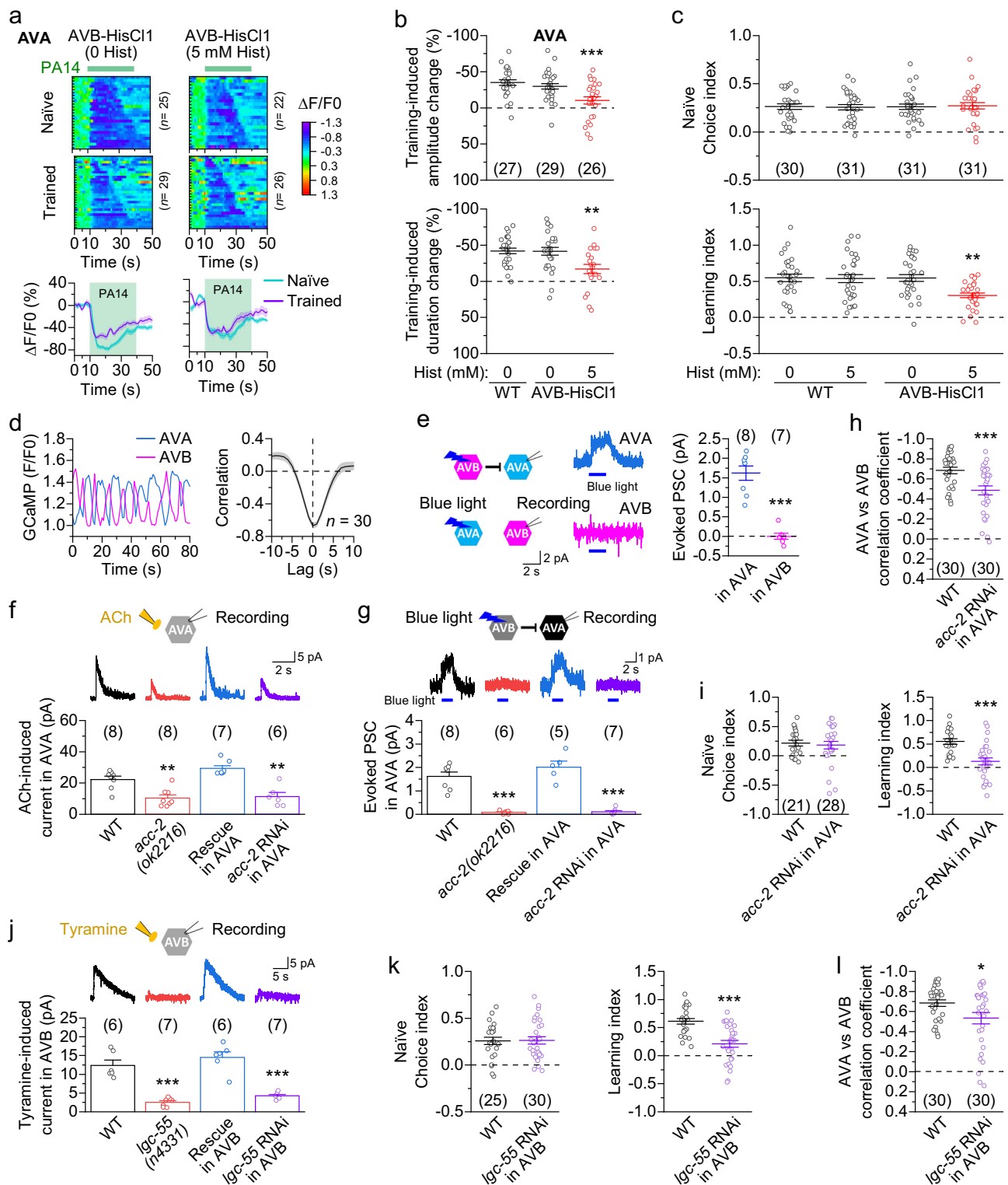

olfactory adaptation to the attractive odorant benzaldehyde[55]. In this assay, we first kept animals on an NGM plate with a drop of benzaldehyde on the lid (Adapted) or without it (Control) for 90 min, and then transferred them to the center of a new NGM plate with a drop of 0.5% benzaldehyde (in ethanol) on one side of the plate and a drop of ethanol on the opposite side (Supplementary Fig. 14a). The numbers of animals at these two spots were counted after 60 min to calculate the chemotaxis index (see equation in Supplementary Fig. 14a). In control groups, the chemotaxis index had positive values of a similar magnitude. In contrast, the chemotaxis index of the adapted wild type was a

large negative value, indicating olfactory adaptation (Supplementary Fig. 14b). The adaptation was partially lost in either *asic-1(ok415)* or *del-1(ok150)* single mutant and completely absent in their double mutant (Supplementary Fig. 14b). These results suggest that proprioception mediated by the mechanoreceptors is also important to olfactory adaptation. Intriguingly, the defect of olfactory adaptation was only partially rescued by A- and B-MN-targeted expression of wild-type *asic-1* and *del-1* (Supplementary Fig. 14b), suggesting that ASIC-1 and DEL-1 also act in other neurons to contribute to the olfactory adaptation.

**Fig. 7 | AVB regulate olfactory learning by interacting with AVA and RIM.**
**a** PA14-evoked Ca$^{2+}$ drop in AVA of naïve and trained animals expressing HisCl1 in AVB (AVB-HisCl1) without and with prior histamine treatment. Top: Heatmaps arranged by the response duration. Bottom: The mean (solid line) and SEM (shaded area). **b** Inhibition of AVB significantly attenuates the PA14 training-induced amplitude and duration changes of the Ca$^{2+}$ drop in AVA. WT = wild type.
**c** Inhibition of AVB significantly decreases the olfactory learning index but not the naïve choice index. **d** Spontaneous GCaMP6 signals of AVA and AVB are anticorrelated. Shown are sample traces recorded simultaneously from AVA and AVB (left), and cross-correlation analysis (right). **e** Optogenetic activation of AVB but not AVA causes an outward (inhibitory) current in its counterpart. ChR2 was expressed in AVB and AVA using *Psra-11* and a *Cre-LoxP* approach (with *Pflp-18* and *Pgpa-14*), respectively. **f** Acetylcholine (ACh)-induced outward current in AVA depends on ACC-2. **g** AVA outward current caused by optogenetic activation of AVB depends on ACC-2. **h** Knockdown of *acc-2* in AVA impairs anticorrelation between AVA and AVB. **i** Knockdown of *acc-2* in AVA greatly reduces the olfactory learning index but not

the naïve choice index. **j** Tyramine-induced outward current in AVB depends on LGC-55. In (**e**–**g**) and **j**, whole-cell currents were recorded at a holding potential of 0 mV (**e**, **g**, **j**) or +30 mV (**f**) using the low-Cl$^-$ pipette solution (see Methods).
**k** Knockdown of *lgc-55* in AVB greatly reduces the olfactory learning index but not the naïve choice index. **l** Knockdown of *lgc-55* in AVB impairs anticorrelation between AVA and AVB. *, **, and *** indicate $p < 0.05$, $p < 0.01$, and $p < 0.001$, respectively, compared with WT based on either one-way ANOVA with Tukey's post hoc test (**b**, **c**, **f**, **g**, **j**) or between two groups based on two-sided unpaired *t* test (**e**, **h**, **i**, **k**, **l**). $p = 0.6343$ and $0.0003$ (top, **b**), $0.9983$ and $0.0037$ (bottom, **b**), $0.9983$, 1, and $0.9974$ (top, **c**), $0.9988$, 1, and $0.0024$ (bottom, **c**), $< 0.0001$ (**e**), $0.0016$, $0.0987$, and $0.0071$ (**f**), $< 0.0001$, $0.3199$, and $< 0.0001$ (**g**), $0.0006$ (**h**), $0.6862$ (left, **i**), $< 0.0001$ (right, **i**), $< 0.0001$, $0.4673$, and $< 0.0001$ (**j**), $0.9223$ (left, **k**), $< 0.0001$ (right, **k**), $0.026$ (**l**). Brackets contain the number of animals tested (**a**, **b**, **d**, **h**, **l**), cells recorded (**e**–**g**, **j**), or independent assays (**c**, **i**, **k**). Data are shown as mean ± SEM. Source data are provided as a Source Data file.

## Discussion

The results of this study demonstrate that locomotion enhances aversive olfactory learning in *C. elegans* by transmitting proprioceptive information from motor neurons to higher-order interneurons. Furthermore, we found that several key elements in the *C. elegans* sensorimotor circuit are necessary for this function of locomotion, including mechanoreceptors in cholinergic motor neurons, antidromic-rectifying gap junctions between locomotion interneurons and motor neurons, and functional interactions among specific interneurons. Based on our findings and existing knowledge, we propose a model depicting how locomotion may contribute to learning (Fig. 8). According to our model, locomotion activates mechanoreceptors in A- and B-MNs, generating proprioceptive information that is transmitted to the locomotion interneurons AVA and AVB through antidromic-rectifying gap junctions. AVA and AVB regulate olfactory learning by interacting with RIM in a disinhibitory circuit, where RIM inhibits AVB, relieving its inhibitory effect on AVA. Previous studies have demonstrated the important roles of AVA, AIB, and RIM in olfactory sensation[50] and olfactory learning[22]. By adding AVB to the olfactory sensorimotor circuit and revealing how it interacts with other neurons to enhance olfactory learning, this study provides major insights into the circuit and synaptic mechanisms of olfactory learning.

We found that proprioception, generated by activating the mechanosensitive channels ASIC-1 and DEL-1 in VA and VB, is necessary for *C. elegans* aversive olfactory learning. The expression of ASIC-1 and DEL-1 in all VA and VB suggests that these channels serve similar functions in all these neurons. However, we cannot disregard the possibility that their physiological roles may differ among neurons of the same type since we have only examined one representative VA and one representative VB. To address this possibility, extensive electrophysiological analyses on multiple motor neurons would be required, which is impractical due to the associated time and cost constraints.

The representatives of DA and DB, DA4 and DB4, also exhibited mechanosensitivity, although the specific mechanoreceptors remain to be identified. Similar to VA and VB, DA and DB connect to AVA and AVB, respectively[19]. This study, along with our previous study[27], indicates that both the AVA/A-MN and AVB/B-MN gap junctions act as strong antidromic rectifiers. While they differ in molecular composition, they maintain identical molecular compositions between a specific pair of locomotion interneurons (AVA or AVB) and their connected motor neurons, regardless of whether the motor neurons innervate ventral or dorsal muscles. As VA and VB proprioception regulates olfactory learning behavior by influencing AVA and AVB activities, it is plausible that DA and DB may also contribute, although their precise functions may differ due to their much smaller sustained MRCs compared to VA and VB and their reliance on other mechanosensory receptors.

In many published models of *C. elegans*, sensory information flows hierarchically from sensory neurons to locomotion interneurons, either directly or indirectly through interneurons, and then to motor neurons, to generate appropriate behavioral responses[56,57]. However, our results indicate that the PA14 training-induced Ca$^{2+}$ drop in AVA, AIB, and RIM interneurons also depends on proprioceptive information from downstream motor neurons, suggesting that interneurons may act as hub neurons to integrate exteroceptive information from olfactory sensory neurons and proprioceptive information from motor neurons to shape olfactory learning. This sensorimotor integration function of interneurons appears to be a common feature of sensorimotor circuits, as it is also observed in *Drosophila*, various insects, and mammals[16,58–60]. Notably, AVA, AIB, and RIM also participate in other learning behaviors such as imprinted olfactory learning and long-term habituation[17,61]. They can also modulate decision-making and probabilistic sensory responses by interacting with other neurons[17,32,50]. Moreover, our results indicate that proprioception also regulates contextual learning. Thus, locomotive proprioception may play a critical role in shaping a variety of learning behaviors in *C. elegans*.

In the *C. elegans* wiring diagram, AVA, AVB, and RIM have many synaptic connections with each other and other neurons[22,30,50]. Notably, RIM have gap junctions and chemical synapses with AIB and AVA, which also have chemical synapses between them (Fig. 8). Although all these interneurons are involved in olfactory learning, it is not completely understood how they interact at the circuit and synaptic levels. Results from a recent study suggest that INX-4, an innexin expressed in RIM, regulates the olfactory learning of PA14 by forming gap junctions[22]. Because INX-4 is only expressed in RIM and not in the gap junction partners AIB and AVA[48,49], the putative gap junctions are likely heterotypic gap junctions. However, the biophysical properties and molecular compositions of these gap junctions are yet to be determined. Furthermore, the molecular identities and functional properties of the postsynaptic receptors that mediate the putative chemical synaptic transmission among RIM, AIB, and AVA are unclear. Therefore, it is challenging to predict how the RIM-AVB-AVA disinhibitory circuit might interact with the other interneurons in shaping olfactory learning.

The PA14-induced Ca$^{2+}$ drop was significantly reduced in wild-type animals after PA14 training. However, this reduction was not observed in animals with impaired motor neuron-mediated proprioception. Our quantitative analyses showed that wild-type animals differed from those with locomotion inhibition or impairment of the motor neuron-mediated proprioception circuit primarily in terms of having a significantly larger PA14-induced Ca$^{2+}$ drop in AVA, AIB, and RIM during the naïve stage, with minimal differences in the post-PA14 training stage. These analyses indicate that the PA14-induced Ca$^{2+}$ drop has both a proprioception-dependent and a proprioception-independent component, and that the occurrence of olfactory learning depends

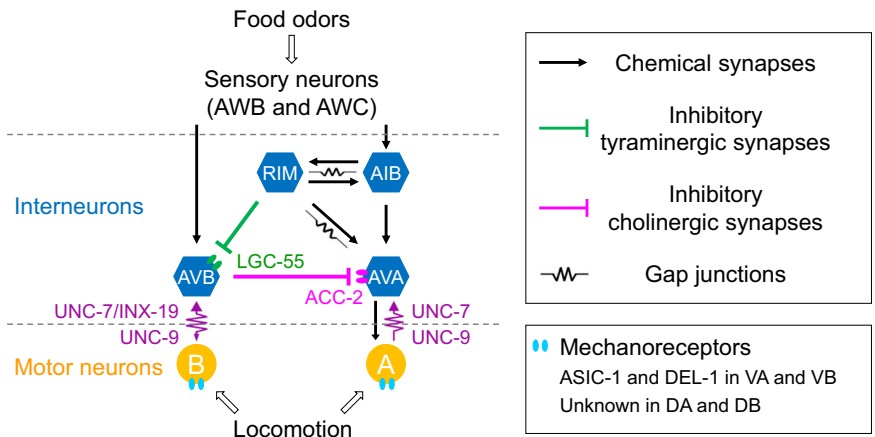

**Fig. 8 | A circuit model for locomotion to regulate olfactory learning through proprioception.** Locomotion activates A- and B-MNs by stimulating their mechanoreceptors, which are formed by ASIC-1 and DEL-1 in VA and VB but unidentified in DA and DB. A- and B-MNs then activate the locomotion interneurons AVA and AVB, respectively, through antidromic-rectifying gap junctions. The gap junctions between AVB and B-MNs are formed by UNC-7 and INX-19 in AVB and UNC-9 in B-MNs, while those between AVA and A-MNs by UNC-7 in AVA and UNC-9 in A-MNs. RIM, AVB, and AVA interact through inhibitory tyraminergic synapses from RIM to AVB, and inhibitory cholinergic synapses from AVB to AVA, with LGC-55 and ACC-2 being the postsynaptic receptors in AVB and AVA, respectively. Gap junctions and excitatory chemical synapses also exist between RIM and AVA.

mainly on whether there is a significant $Ca^{2+}$ drop between the naïve and post-PA14 training stages. Consistently, a mutation of *nmr-1*, which encodes the *C. elegans* homolog of the NR1 subunit of mammalian NMDA receptors, also impairs PA14 olfactory learning and affects the PA14-induced $Ca^{2+}$ drop in AVA, AIB, and RIM mainly during the naïve stage[22]. One question that has arisen from our analyses of the $Ca^{2+}$ drop is why inhibition of the motor neuron-mediated proprioception circuit did not alter the naïve choice index. This is likely because the naïve preference of *C. elegans* for PA14 is determined by the intrinsic properties of AWB and AWC sensory neurons[21]. However, these sensory neurons do not receive synaptic inputs from either A- and B-MNs or the AVA, AIB, RIM, and AVB interneurons[30].

Learning is a complex neurological process that involves multiple neural circuits and can result in diverse behaviors depending on the types of sensory inputs[21–24]. This study demonstrates that locomotion can facilitate aversive olfactory learning in *C. elegans* by transmitting proprioceptive information from motor neurons to locomotion interneurons, which in turn interact with other interneurons to shape the olfactory learning behavior. Interestingly, human patients with mutations of the mechanosensitive channel PIEZO2, the principal stretch-sensitive channel for proprioception[62], often exhibit severe cognitive impairment manifested by various symptoms such as limited vocabulary learning abilities[63,64]. The apparent connection between deficiencies in mechanoreceptors and cognitive impairment in humans suggests a potential link between proprioception and learning capacity. Furthermore, the aversive olfactory learning circuit in *C. elegans* shares major functional and structural similarities with sensorimotor circuits in other species. Specifically, it conveys locomotor information as proprioception to higher-order neurons[16], integrates sensory and motor information through interneurons[16,58,60], and incorporates a disinhibitory circuit[7,65–67]. These similarities imply that the mechanistic insights gained from this study could have broader implications for understanding how locomotion contributes to learning across different species.

## Methods

### *C. elegans* strains and culture
All animals were raised on Nematode Growth Medium (NGM) plates seeded with *E. Coli* OP50 at 22 °C. The strains used in this study are listed in Table 1. Transgenic strains were generated by injecting plasmid DNA. Integrated transgenic strains were generated by either gamma or UV irradiation, and outcrossed 3 times before use. All experiments were performed with young adult hermaphrodites.

### Gene expression pattern analyses
The expression patterns of *asic-1* and *del-1* were assessed by expressing GFP under the control of a 3.7-kb *Pasic-1* and mStrawberry under the control of a 2-kb *Pdel-1*, respectively. The ventral nerve cord motor neurons expressing *asic-1* or del-1 were identified based on their anatomical locations. Other neurons expressing *asic-1* were identified based on the detection of both green and red fluorescent signals in animals that co-expressed GFP under the control of *Pasic-1* and mStrawberry under the control of *Pdat-1* (CEP, ADE, and PDE)[68], *Pegl-46* (FLP)[69], or *Pser-2prom3* (PVD)[70]. The co-expression of *asic-1* and *del-1* in VB and VA was confirmed by co-expressing *Pasic-1::GFP* and *Pdel-1::mStrawberry* as transgenes. All promoters were cloned using the genomic DNA of the Bristol N2 strain as the template and the primers listed in Supplementary Table 1. Images were captured using an inverted microscope (Ts2R, Nikon) equipped with a CCD camera (MD60, Mshot).

### Mutant rescue and RNAi
Mutants were rescued by expressing wild-type cDNAs. Knockdown (RNAi) was achieved by co-expressing two plasmids encoding complementary sense and antisense mRNA fragments of the target gene. The promoters used for cell-specific rescue and RNAi included *Punc-17Δ1* (ventral cord cholinergic motor neurons)[38], *Pdel-1* (VA and VB), *Psra-11* (AVB)[28,32], and *Pacr-5* (B-MNs)[28,32]. AVA-specific expression was achieved by using a *Cre-LoxP* approach involving *Pflp-18* and *Pgpa-14*[27,32]. All cDNAs and target mRNA fragments were cloned from a Bristol N2 cDNA library using the primers listed in Supplementary Table 1.

### Electrophysiology
Electrophysiological experiments were performed as previously described[32]. Briefly, a young adult hermaphrodite was glued on a glass coverslip in a solution identical to the bath solution used in electrophysiological experiments. A longitudinal incision was made along the glued region to expose neurons of interest. After clearing the viscera, the cuticle flap was folded back and glued to the coverslip. The dissected preparation was treated with collagenase A (Roche Applied Science, catalog number 10103578001) at a concentration of 0.5 mg

**Table 1 | List of *C. elegans* strains**

| Genotype | Source | Strain ID |
|---|---|---|
| Wild-type Bristol | CGC | N2 |
| *asic-1(ok415)* | CGC | RB680 |
| *del-1(ok150)* | CGC | NC279 |
| *acd-2(ok1237)* | CGC | RB1192 |
| *acd-3(ok1335)* | CGC | VC1047 |
| *acd-4(ok1508)* | CGC | RB1351 |
| *acd-5(ok2657)* | CGC | RB2005 |
| *asic-2(ok289)* | CGC | RB557 |
| *deg-1(u38)* | CGC | TU38 |
| *degt-1(ok3307)* | CGC | VC2633 |
| *del-3(ok2613)* | CGC | RB1979 |
| *del-4(ok1014)* | CGC | RB1064 |
| *del-7(ok1187)* | CGC | RB1156 |
| *del-8(ok1357)* | CGC | VC831 |
| *del-9(ok2353)* | CGC | RB1818 |
| *del-10(ok1705)* | CGC | RB1469 |
| *delm-1(ok1226)* | CGC | RB1177 |
| *delm-2(ok1822)* | CGC | RB1523 |
| *egas-1(ok3497)* | CGC | RB2521 |
| *egas-2(ok1477)* | CGC | VC975 |
| *egas-3(ok1522)* | CGC | RB1356 |
| *flr-1(ut11)* | CGC | JC55 |
| *mec-4(u253)* | CGC | TU253 |
| *mec-10(tm1552)* | CGC | ZB2551 |
| *unc-8(tm2071)* | Miriam B. Goodman | ZW1013 |
| *unc-105(ok1432)* | CGC | RB1316 |
| *asic-1(ok415);del-1(ok150)* | This paper | PLX214 |
| *asic-1(ok415);xyhEx425[Punc-17Δ1::asic-1(cDNA), Pmyo-2::mStrawberry]* | This paper | PLX425 |
| *del-1(ok150);xyhEX270[Punc-17Δ1::del-1(cDNA)::SL2::GFP, Pmyo-2::mStrawberry]* | This paper | PLX270 |
| *asic-1(ok415);del-1(ok150);xyhEx151[Punc-17Δ1::asic-1(cDNA)::SL2::mStrawberry, Punc-17Δ1::del-1(cDNA)::SL2::mStrawberry, Pmyo-2::GFP]* | This paper | PLX151 |
| *xyhIs307[Pmyo-3::HisCl1::SL2::GFP, Pmyo-2::mStrawberry]* | This paper | PLX307 |
| *xyhEx39[Ptdc-1::GCaMP6s, Pnpr-9::GCaMP6s, Pflp-18::loxP::LacZ::STOP::loxP::GCaMP6s, Pgpa-14::Cre, lin-15(+)]* | This paper | PLX39 |
| *xyhEx228[Pmyo-3::HisCl1::mStrawberry; Punc-122::dsRed];xyhEx39[Ptdc-1::GCaMP6s, Pnpr-9::GCaMP6s, Pflp-18::loxP::LacZ::STOP::loxP::GCaMP6s, Pgpa-14::Cre, lin-15(+)]* | This paper | PLX228 |
| *xyhEx192[Pasic-1::GFP, Pdel-1::SL2::mStrawberry]* | This paper | PLX192 |
| *xyhEx238[Pser-2prom3::GFP, Pasic-1::mStrawberry]* | This paper | PLX238 |
| *xyhEx239[Pdat-1::mStrawberry, Pasic-1::GFP]* | This paper | PLX239 |
| *xyhEx240[Pegl-46::mStrawberry, Pasic-1::GFP]* | This paper | PLX240 |
| *asic-1(ok415);del-1(ok150);xyhEx39[Ptdc-1::GCaMP6s, Pnpr-9::GCaMP6s, Pflp-18::loxP::LacZ::STOP::loxP::GCaMP6s, Pgpa-14::Cre, lin-15(+)]* | This paper | PLX197 |
| *asic-1(ok415);del-1(ok150);xyhEx39[Ptdc-1::GCaMP6s, Pnpr-9::GCaMP6s, Pflp-18::loxP::LacZ::STOP::loxP::GCaMP6s, Pgpa-14::Cre, lin-15(+)];xyhEx303[Pdel-1::SL2::asic-1(cDNA):: SL2::mStrawberry, Pdel-1::SL2::del-1(cDNA), Punc-122::dsRed]* | This paper | PLX303 |
| *asic-1(ok415);del-1(ok150);xyhEx306[Pdel-1::SL2::del-1(cDNA), Pdel-1::SL2::asic-1(cDNA)::SL2::mStrawberry, Pmyo-2::GFP]* | This paper | PLX306 |
| *asic-1(ok415);del-1(ok150);xyhIs307[Pmyo-3::HisCl1::SL2::GFP, Pmyo-2::mStrawberry]* | This paper | PLX310 |
| *xyhIs24[Psra-11::GFP]* | This paper | PLX24 |
| *inx-1(tm3524);zwIs142[Psra-11::GFP]* | This paper | ZW230 |
| *zwEx286[Punc-17Δ1::: inx-3ss, Punc-17Δ1:: inx-3as];zwIs142[Psra-11::GFP(wp712)]* | This paper | ZW1452 |
| *inx-7(tm2738);zwIs142[Psra-11::GFP]* | This paper | ZW1314 |
| *inx-10(ok2714);zwIs142[Psra-11::GFP]* | This paper | ZW231 |
| *zwEx287[Punc-17Δ1::: inx-12ss, Punc-17Δ1:: inx-12as];zwIs142[Psra-11::GFP]* | This paper | ZW1458 |
| *inx-14(ag17);zwIs142[Psra-11::GFP]* | This paper | ZW1315 |
| *unc-7(e5);xyhIs24[Psra-11::GFP]* | This paper | PLX273 |
| *unc-9(fc16);xyhIs24[Psra-11::GFP]* | This paper | PLX274 |
| *inx-19(tm1896);xyhEx42[Psra-11::GFP]* | This paper | PLX126 |
| *unc-7(e5);inx-19(tm1896);xyhEx42[Psra-11::GFP]* | This paper | PLX42 |

**Table 1 (continued) | List of C. elegans strains**

| Genotype | Source | Strain ID |
|---|---|---|
| xyhEx112[Psra-11::unc-7ss, Psra-11::unc-7as, Psra-11::GFP] | This paper | PLX112 |
| xyhEx370[Pacr-5::unc-9ss, Pacr-5::unc-9as, Psra-11::GFP] | This paper | PLX370 |
| xyhEx155[Psra-11::inx-19ss, Psra-11::inx-19as, Psra-11::GFP] | This paper | PLX155 |
| xyhEx118[Pacr-11::unc-7ss, Pacr-11::unc-7as, Pasr-11::inx-19ss, Pasr-11::inx-19as, Pmyo-2::GFP] | This paper | PLX118 |
| xyhEx373[Psra-11::unc-7ss, Psra-11::unc-7as, Psra-11::inx-19ss, Psra-11::inx-19as, Psra-11::GFP] | This paper | PLX373 |
| zwEx175[Pflp-18::loxP::LacZ::STOP::loxP::mCherry::SL2::GFP, Pgpa-14::Cre] | This paper | ZW704 |
| unc-7(e5);zwEx175[Pflp-18::loxP::LacZ::STOP::loxP::mCherry::SL2::GFP, Pgpa-14::Cre] | This paper | ZW731 |
| xyhEx203[Pflp-18::loxP::LacZ::STOP::loxP::unc-7ss, Pflp-18::loxP::LacZ::STOP::loxP::unc-7as, Pgpa-14::Cre, Pmyo-2::GFP] | This paper | PLX203 |
| zwEX202[Pflp-18::loxP::LacZ::STOP::loxP::unc-7ss, Pflp-18::loxP::LacZ::STOP::loxP::unc-7as, Pgpa-14::Cre, Pmyo-2:: mStrawerry];zwEx175[Pflp-18::loxP::LacZ::STOP::loxP::mCherry::SL2::GFP, Pgpa-14::Cre] | This paper | ZW933 |
| xyhEx39[Ptdc-1::GCaMP6s, Pnpr-9::GCaMP6s, Pflp-18::loxP::LacZ::STOP::loxP::GCaMP6s, Pgpa-14::Cre, lin-15(+)];xyhEx449[Psra-11::unc-7ss, Psra-11::unc-7as, Psra-11::inx-19ss, Psra-11::inx-19as, Pflp-18::loxP::LacZ::STOP::loxP::unc-7ss, Pflp-18::loxP::LacZ::STOP::loxP::unc-7as, Pgpa-14::Cre, Punc-122::dsRed] | This paper | PLX449 |
| xyhIs307[Pmyo-3::HisCl1::SL2::GFP, Pmyo-2::mStrawberry];xyhEx454[Psra-11::unc-7ss, Psra-11::unc-7as, Psra-11::inx-19ss, Psra-11::inx-19as, Pflp-18::loxP::LacZ::STOP::loxP::unc-7ss, Pflp-18::loxP::LacZ::STOP::loxP::unc-7as, Pgpa-14::Cre, Pmyo-2::GFP] | This paper | PLX454 |
| xyhEx200[Psra-11::HisCl1::SL2::mStrawberry, Punc-122::dsRed];xyhEx39[Ptdc-1::GCaMP6s, Pnpr-9::GCaMP6s, Pflp-18::loxP::LacZ::STOP::loxP::GCaMP6s, Pgpa-14::Cre, lin-15(+)] | This paper | PLX200 |
| xyhIs281[Psra-11::GCaMP6s, Pflp-18::loxP::LacZ::STOP::loxP::GCaMP6s, Pgpa-14::Cre] | This paper | PLX281 |
| kyEx3801[Psra-11::ChR2::GFP, Punc-122::dsRed];zwIs143[Pflp-18::loxP::LacZ::STOP::loxP::mStrawberry, Pgpa-14::Cre, lin-15(+)] | This paper | ZW1191 |
| ntIs[Prig-3::ChR2, Punc-122::dsRed];ntIs35[Psra-11::tdTomato];lite-1(ce314) | Shawn R. Lockery | XL238 |
| acc-1(tm3268);zwEx175[Pflp-18::loxP::LacZ::STOP::loxP::mCherry::SL2::GFP, Pgpa-14::Cre] | This paper | ZW1185 |
| acc-2(ok2216);zwEx175[Pflp-18::loxP::LacZ::STOP::loxP::mCherry::SL2::GFP, Pgpa-14::Cre] | This paper | ZW1186 |
| acc-3(ok3450);zwEx175[Pflp-18::loxP::LacZ::STOP::loxP::mCherry::SL2::GFP, Pgpa-14::Cre] | This paper | ZW1183 |
| acc-4(ok2371);zwEx175[Pflp-18::loxP::LacZ::STOP::loxP::mCherry::SL2::GFP, Pgpa-14::Cre] | This paper | ZW1184 |
| lgc-46(ok2949);zwEx175[Pflp-18::loxP::LacZ::STOP::loxP::mCherry::SL2::GFP, Pgpa-14::Cre] | This paper | ZW1215 |
| lgc-47(ok2963);zwEx175[Pflp-18::loxP::LacZ::STOP::loxP::mCherry::SL2::GFP, Pgpa-14::Cre] | This paper | ZW1220 |
| lgc-48(gk964294);zwEx175[Pflp-18::loxP::LacZ::STOP::loxP::mCherry::SL2::GFP, Pgpa-14::Cre] | This paper | ZW1229 |
| lgc-49(tm6556);zwEx175[Pflp-18::loxP::LacZ::STOP::loxP::mCherry::SL2::GFP, Pgpa-14::Cre] | This paper | ZW1181 |
| acc-2(ok2216);kyEx3801[Psra-11::ChR2::GFP, Punc-122::dsRed];zwIs143[Pflp-18::loxP::LacZ::STOP::loxP::mStrawberry, Pgpa-14::Cre, lin-15(+)] | This paper | ZW1196 |
| acc-2(ok2216);kyEx3801[Psra-11::ChR2::GFP, Punc-122::dsRed];zwEx277[Pflp-18::loxP::LacZ::STOP::loxP::acc-2(cDNA), Pflp-18::loxP::LacZ::STOP::loxP::mCherry::SL2::GFP, Pgpa-14::Cre] | This paper | ZW1450 |
| kyEx3801[Psra-11::ChR2::GFP, Punc-122::dsRed];zwEx284[Pflp-18::loxP::LacZ::STOP::loxP::acc-2 ss;Pflp-18::loxP::LacZ::STOP::loxP::acc-2 as;Pflp-18::loxP::LacZ::STOP::loxP::mCherry::SL2::GFP, Pgpa-14::Cre] | This paper | ZW1469 |
| xyhEx264[Pflp-18::loxP::LacZ::STOP::loxP::acc-2ss, Pflp-18::loxP::LacZ::STOP::loxP::acc-2as, Pgpa-14::Cre, Pmyo-2::GFP] | This paper | PLX264 |
| xyhIs281[Psra-11::GCaMP6s, Pflp-18::loxP::LacZ::STOP::loxP::GCaMP6s, Pgpa-14::Cre, Punc-122::deRed];xyhEx285[Pflp-18::loxP::LacZ::STOP::loxP::acc-2ss, Pflp-18::loxP::LacZ::STOP::loxP::acc-2as, Punc-122::dsRed] | This paper | PLX285 |
| lgc-55(n4331);xyhIs24[Psra-11::GFP] | This paper | PLX84 |
| lgc-55(n4331);xyhEx275[Psra-11::lgc-55(cDNA), Psra-11::GFP, Punc-122::dsRed] | This paper | PLX275 |
| xyhEx278[Psra-11::lgc-55ss, Psra-11::lgc-55as, Psra-11::GFP, Pmyo-2::GFP] | This paper | PLX278 |
| xyhEx267[Psra-11::lgc-55ss, Psra-11::lgc-55as, Pmyo-2::GFP] | This paper | PLX267 |
| xyhIs281[Psra-11::GCaMP6s, Pflp-18::loxP::LacZ::STOP::loxP::GCaMP6s, Pgpa-14::Cre];xyhEx288[Psra-11::lgc-55ss, Psra-11::lgc-55as, Punc-122::dsRed] | This paper | PLX288 |
| xyhEx348[Psra-6::GCaMP6s, Punc-122::dsRed] | This paper | PLX348 |
| asic-1(ok415);del-1(ok150);xyhEx348[Psra-6(2.4kb)::GCaMP6s, Punc-122::dsRed] | This paper | PLX455 |
| xyhEx456[Pmyo-3::HisCl1::SL2::mStrawberry, Psra-6::GCaMP6s, Punc-122::GFP] | This paper | PLX456 |
| xyhEx348[Psra-6::GCaMP6s, Punc-122::dsRed];xyhEx459[Psra-11::unc-7ss, Psra-11::unc-7as, Psra-11::inx-19ss, Psra-11::inx-19as, Pflp-18::loxP::LacZ::STOP::loxP::unc-7ss, Pflp-18::loxP::LacZ::STOP::loxP::unc-7as, Pgpa-14::Cre, Punc-122::GFP] | This paper | PLX459 |

ml$^{-1}$ for 10-15 s and perfused with the bath solution. Borosilicate glass pipettes (tip resistance ~20 MΩ) were used as electrodes. A classical whole-cell configuration was obtained by applying negative pressure to the recording pipette. Motor neurons were identified based on their anatomical locations. Identification of AVB and AVA was aided by fluorescent labeling using *Psra-11*[32] and the *Cre-LoxP* approach involving *Pflp-18* and *Pgpa-14*[27,32], respectively. Voltage- and current-clamp recordings were performed on a Nikon FN1 microscope with a Multi-Clamp 700B amplifier (Molecular Devices) and the Clampex software (version 10, Molecular Devices). The holding potential was −60 mV

unless when specified otherwise. Pressure ejections of exogenous ACh (1 mM, ACROS Organics, catalog number AC159170050) and tyramine (200 μM, Sigma-Aldrich, catalog number T2879), diluted to their final concentrations in the bath solution, were done using a Picospritzer III microinjector (Parker Hannifin) through a glass pipette (tip diameter ~2 μm) at 2 psi. Mechanical stimuli were delivered by pressure-ejecting the bath solution using the Picospritzer III microinjector.

The bath solution contained (in mM) 140 NaCl, 5 KCl, 5 CaCl$_2$, 5 MgCl$_2$, 11 dextrose, and 5 HEPES (pH 7.2). To assay Ca$^{2+}$ permeability, CaCl$_2$ in the bath solution was completely removed (Ca$^{2+}$-free) or

replaced by 5 mM $BaCl_2$. To assay $Na^+$ permeability, 136 mM NaCl in the bath solution was replaced ($Na^+$-free) with an equal molar concentration of choline chloride or NMDG. To block $Na^+$ channels, amiloride (300 μM, Sigma-Aldrich, catalog number A7410) was added to the bath solution. The standard pipette solution contained (in mM) 120 KCl, 20 KOH, 5 Tris, 0.25 $CaCl_2$, 4 $MgCl_2$, 36 sucrose, 5 EGTA, and 4 $Na_2ATP$ (pH 7.2). A low-$Cl^-$ pipette solution containing (in mM) 6.8 KCl, 113.2 Kgluconate, 20 KOH, 5 Tris, 0.25 $CaCl_2$, 4 $MgCl_2$, 36 sucrose, 5 EGTA, and 4 $Na_2ATP$ (pH 7.2) was used in some experiments as specified in figure legends.

## Optogenetic stimulation

Transgenic animals expressing channelrhodopsin-2 (ChR2) were first grown to the L1-L2 stage on an NGM plate, and then transferred to a new NGM plate with all-trans-retinal (Sigma-Aldrich, catalog number R2500) two days before the electrophysiological experiment. The retinal plate was prepared by spotting a 6-cm petri dish containing 10 ml NGM with 200 μl *E. coli* OP50 supplemented by 2 mM all-trans-retinal. Blue light pulses were generated by a Lambda XL light source (Sutter Instrument) with a 470 ± 20 nm excitation filter (59222, Chroma Technology Corp.) and applied through a 40X water immersion objective. The measured light intensity at the specimen position was 4.38 mW mm$^{-2}$. The on-and-off of light stimulation was controlled by the NIS-Elements imaging software (version 4.51, Nikon).

## $Ca^{2+}$ imaging

Young adult transgenic animals expressing GCaMP6 were partially restrained by gluing only the head region for imaging $Ca^{2+}$ signals in head neurons. During spontaneous GCaMP6 recordings, animals were maintained in the bath solution. For experiments assessing the effects of bacterial supernatants on $Ca^{2+}$ signal, PA14 and OP50 were cultured in LB medium overnight at 26 °C and 37 °C, respectively. Bacterial supernatant was collected by two rounds of centrifugation at 1520g for 20 min followed by filtering (0.22 μm). During recordings, animals were maintained in LB, and the bacterial supernatant was applied to the vicinity of the nose from the side (at a 90-degree angle) through pressure ejections (2 psi, 30 s) using the Eppendorf FemtoJet 4i microinjector. $Ca^{2+}$ imaging was collected for 3 min at 1 frame per second on an inverted microscope (Ts2R, Nikon) equipped with a CCD camera (MD60, Mshot) using MShot Image Analysis System (version 1.1, Mshot).

## Behavioral assays

Locomotion behavior was analyzed at 22 °C. An automated *Track-A-Worm* system was used for imaging acquisition and subsequent quantitative analyses[34]. Briefly, a single young adult hermaphrodite was transferred to a 6-cm NGM plate without bacteria. After a 30-s recovery time from the transfer, the animal was imaged for 1 min at 15 frames per second.

Aversive olfactory learning assays were performed at 22 °C following a protocol modified from previous studies[17,23]. Briefly, a cohort of synchronized animals were grown on *E. coli* OP50 until adulthood, and then transferred to two training plates covered by a fresh lawn of OP50 (naïve control) and PA14 (aversive training), respectively. After a 6-h training, 5 animals from each training plate were randomly picked to examine their locomotion speed. The remaining animals were washed off with M9 buffer, rinsed twice, and placed in the middle of a 9-cm choice plate (100-150 animals per plate) to test their olfactory preferences between OP50 and PA14. After 1 h, animals at each spot were counted (Fig. 1a). The training plates containing histamine (Sigma-Aldrich, catalog number H7250) or gelatin (Shanghai Aladdin Biochemical Technology Co., Ltd, catalog number G108395) were prepared by first adding their final concentration to the solution for making the NGM plates, and then spotting each plate with 200 μl bacteria containing histamine or gelatin of the same concentration. In the choice test, 30 μl fresh overnight culture of OP50 and PA14 (diluted to $OD_{600}$ = 1) was applied to opposite sides of a 9-cm NGM plate. The choice index and learning index were calculated using the equations shown in Fig. 1a, which were based on previous studies[17,22,32].

Olfactory adaptation assays were performed at 22 °C[55]. Briefly, animals were incubated without (control) or with (adapted) 100% benzaldehyde applied on the lid of the NGM plate (2 μl, Sigma-Aldrich, catalog number B1334) for 90 min, then washed twice with M9 buffer and placed in the middle of a 6-cm choice plate (~50 animals per plate) to test their olfactory preferences between 1:200 benzaldehyde:EtOH (1 μl) and the diluent EtOH (1 μl), which were placed on opposite sides of the plate. 1 μl of 0.05 M sodium azide was added to each spot to immobilize the animals. After 60 min, animals at each spot were counted, and a chemotaxis index was calculated (Supplementary Fig. 14a).

## Data analyses

Electrophysiological data were quantified using Clampfit (version 10, Molecular Devices). Junctional currents ($I_j$) were recorded from neuron #1, which was held constantly at a membrane voltage ($V_m$) of −30 mV, while a series of $V_m$ steps were applied to neuron #2 from a holding voltage of −30 mV. Transjunctional voltage ($V_j$) = $V_m$ of neuron #1 (−30 mV) - $V_m$ of neuron #2. Amplitudes of $I_j$ and sustained MRCs of motor neurons were quantified from the mean current during the last 100 ms of a $V_j$ step or a mechanical stimulus. Amplitudes of inward current and membrane depolarization in AVA and AVB caused by mechanical stimulation of motor neurons, and amplitudes of optogenetically evoked PSCs were quantified from the mean current or membrane potential during the stimulus.

$Ca^{2+}$ imaging data were analyzed using ImageJ (version 1.50i, National Institutes of Health) by assigning individual cells as separate regions of interest (ROI). For spontaneous $Ca^{2+}$ signals, fluorescence intensities (F) in each ROI were first plotted as absolute intensities over time, and then converted to a bleach-corrected F/F0 graph using MATLAB (version R2023a, MathWorks) running a custom module, which generates the corrected baseline and F/F0 peaks using a cubic interpolation algorithm (see Code Availability for further information). F/F0 events that were ≥0.1 above the baseline and lasting ≥3 s were used for quantifying the frequency and amplitude of $Ca^{2+}$ peaks. The pairwise cross-correlation of spontaneous $Ca^{2+}$ transients between neurons was calculated using a Correlations Coefficients function in Origin (version 2019, OriginLab). For PA14- or OP50-stimulated $Ca^{2+}$ imaging, fluorescence intensity (F) in each ROI was generated by subtracting the background intensity (F0) from the total intensity. ΔF/F0 at each time point was calculated using the equation (F-F0)/F0, where F0 is the mean value of 10 s before the stimulation. The amplitude of the evoked response was then determined by calculating the mean value of the first 20 s during the stimulus. The training-induced amplitude change of $Ca^{2+}$ drop was calculated using the formula ((amplitude of trained animals/amplitude of naïve animals)−1) × 100%. The duration of OP50- or PA15-induced $Ca^{2+}$ drop was measured from the occurrence of the sharp drop in ΔF/F0 to the moment when ≥70% of the $Ca^{2+}$ drop had perished. The training-induced duration change of $Ca^{2+}$ drop was calculated using the formula ((duration of trained animals/duration of naïve animals)−1) × 100%.

Statistical analyses and data graphing were performed with Origin (version 2019, OriginLab). All data are shown as mean ± SEM. one-way ANOVA with Tukey's post hoc test, two-sided unpaired *t* test, or two-sided Fit comparison with *F*-test was used for statistical comparisons as specified in figure legends.

## Reporting summary

Further information on research design is available in the Nature Portfolio Reporting Summary linked to this article.

## Data availability

All data generated or analyzed during this study are included in this published article and its supplementary information files. Source data are provided with this paper.

## Code availability

The MATLAB module used to correct the photobleaching-induced drop of the fluorescence signal, along with user instructions, is freely available at https://health.uconn.edu/worm-lab/track-a-worm/.

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

## Acknowledgements

This work was supported by grants from the National Natural Science Foundation of China (32171003 to P.L. and 81803917 to C.C.) and the National Institute of Health (R01NS109388 and R01MH085927 to Z.-W.W.). We thank Shawn R. Lockery for the XL238 strain, Miriam B. Goodman for *unc-8(tm2071)*, Daniel A. Colon-Ramos for the *Punc-122::dsRed* plasmid, Cori Bargmann for a HisCl1 plasmid, Xiang Yang for a GCaMP6s plasmid, and the Caenorhabditis Genetics Center (USA), which is funded by NIH Office of Research Infrastructure Programs (P40 OD010440), for other mutant strains.

## Author contributions

X.Z., Y.L., X.D., and R.D. performed molecular biological, genetic, and behavioral experiments. P.L. conducted all the Ca2+ imaging and the majority of electrophysiological experiments. L.N. performed some electrophysiological experiments. X.Z., C.C., Z.-W.W., and P.L. analyzed and interpreted the results. P.L. and C.C. designed the research. P.L. and Z.-W.W. wrote the manuscript.

## Competing interests

The authors declare no competing interests.
