## [Peer Review File · Nature Communications]

Locomotion modulates olfactory learning through proprioception in *C. elegans*REVIEWER COMMENTS

Reviewer #1 (Remarks to the Author):

In Zhan et al. manuscript, the authors set out to address an interesting question on brain functions: how locomotion modulates learning via proprioception. They addressed this question by examining the modulation of three pairs of interneurons AVA, AIB and RIM by locomotion in *C. elegans*. These 3 pairs of neurons are connected with chemical synapses and gap junctions and play important roles in olfactory chemotaxis and its plasticity. Learning-generated changes in the activity and coupling of AVA, AIB and RIM are critical for learned olfactory chemotaxis. They first showed that inhibiting body wall muscles disabled learning and decreased the activities and coupling of AVA, AIB and RIM. To understand how locomotion regulates learning and the activities of AVA, AIB and RIM, they found that ASIC-1 and DEL-1 regulated the response of VB6 and VA5 motor neurons to pressure/mechanical stimulation, naïve and trained activity of AVA, and the enhancement of learning by locomotion. They further showed that VB6 and VA5 motor neurons transmitted information of pressure to AVB and AVA interneurons via gap junctions that were also required for learning. AVB regulate AVA via *acc-2* and respond to tyramine signals via *lgc-55*; these regulations are required for learning. The main contributions of the study include demonstrating the dependence of olfactory learning on locomotion (represented by BWM activity, and mechanical responses in VA and VB motor neurons), mapping the circuit that transmits the locomotory information to interneurons that integrate sensory-motor information to regulate learning, and characterizing the properties of VA and VB motor neurons, and ASIC-1 and DEL-1 expressed in these neurons, as proprioceptive receptors. These findings are new. As a whole, the experimental design is suitable for the question under investigation and the results are compelling.

My main question is on the characterization of VB6 and VA5 motor neurons as proprioceptive receptors. This is the key contribution of the study. They characterized these neurons using “The mechanostimulation was achieved by pressure-ejecting (10 psi, 300 ms)...”. The idea is to use this stimulation pattern to mimic the pressure and stretches generated by the locomotion of the worm. Are the choice of the pressure and duration physiologically reasonable/relevant? This needs to be justified. Are these neurons responsive to a range of pressures or durations? A more in-depth characterization of this question would strength the overall contribution of this study.

A second question is about the speed-dependency of learning. It is quite intriguing to see an almost linear effect. Does this imply that locomotion of a lower/higher speed (as a result of more/less BWM inhibition) generates weaker/stronger response in VB6 and VA5 motor neurons and subsequently different levels of regulation of AVB and AVA? Does a lower speed in locomotion indicate weaker pressures delivered to the VB6 and VA5 motor neurons during movement or a lower frequency of the mechanical stimulation generated by the movement? If feasible, some experimental tests would be certainly great. At least some discussion and clarifications will be helpful.

Minor points:

1. Fig3e,f, it seems that each mutant was used in two types of comparisons – these comparisons and

corrections need to be clearly labeled.

2. The authors have tested and characterized VB6 and VA5 motor neurons as sensors for mechanical stimuli. Does this lead to the conclusion that other VA and VB neurons are also stretch sensors? The authors may be right, but I suggest using some caution in this generalization.

3. Fig4e, was any statistical test performed?

Reviewer #2 (Remarks to the Author):

In their manuscript “Locomotion modulates olfactory learning through proprioception in *C. elegans*” the authors follow an intriguing finding that proprioception is involved in aversive learning, to discover the underlying circuit and molecular mechanisms. On that path, they also discover unknown details about the locomotion circuit. The authors use an impressive range of well-chosen methods, from genetics and transgenics, through optogenetic recording and simulation, chemogenetic hyperpolarization, to electrophysiology and behavior analysis. Their approach is thoughtful, though, and methodical. The study is timely, and it provides both specific and generalizable insights that should be of interest to researchers of several fields.

I have a few concerns and suggestions:

Major comments

Across different sections the authors interchange exercise and locomotion. It is not clear whether they suggest that generic coincidental exercise improves learning capabilities (suggested by words like “beneficial” and the human-related introduction), or that proprioceptive input is integrated with exteroceptive input to give embodiment, context, and emphasis to the memory. I find the latter much more exciting and worthwhile. The duration and strength of all proprioceptive inputs might serve, for example, as a proxy for overall responsiveness. The vertebrate cerebellum and other examples (given in the discussion) could indeed have similar roles in motor learning; their circuit motif similarities only add to this hypothesis. I suggest rewriting to reduce the “generic coincidental exercise” parts and clarify the “proprioceptive-exteroceptive integration” parts.

In their narrative and conclusion (including the model in fig. 7) the authors leave out parts of the circuit and focus, without clear justification, on other parts. For example, they describe AVB as the main conduit of ascending input while similar proprioceptive input is mediated by AVA that has similar rectifying GJ with VA. Similarly, RIM directly connects to AVA in addition to the disinhibition via AVB. Finally, proprioceptive MRC in dorsal MN exist even if the mechanoreceptor was not identified; input from DB is demonstrated to be mediated through *unc-7* and *inx-19*, but not for DA and not whether they are paired with *unc-9*. Each of these parts should either be included in the narrative and conclusion, or its exclusion should be justified (or justified more clearly).

Related issue that was “left behind” is that it seems that *inx-19* has another partner in AVB that is not

unc-9 (because unc-9(ko) recapitulate unc-7(ko), not the double mutant L329). It could be nsy-5 (see Altun et al 2009). If nsy-5(ko) exists, the experiments will complete the knowledge. If ko not accessible, address it like the 5 untested mechanoreceptors, by pointing it out and writing that it is not available.

How did the author restrict the mechanical stimulation to those neurons? For example, when stimulating either VB6 or VA5 in Fig 5 and L344, how did they aim for one or the other. Similarly, when recording from a stimulated MN in Fig. 3, how did the author exclude mechanical stimulation of neighboring MN that have synaptic connections to the recorded MN (for example, the soma of VB6 overlaps in the ventral nerve cord with the NMJ area of VB5 and the asynaptic neurite that extends after the axons of VB1-4)?

The dorsal MN produced MRC without a sustained current (Fig. 3) compared to the ventral MNs. Did the author stimulate the commissure and dorsal cord portions of these MNs?

The causality of AVA-AVB anticorrelation (e.g. L405) and learning is not supported. They could each be caused by proprioception that was demonstrated necessary for both.

Minor comments:

(More comments and edits are also on attached PDF file)

L656-657 Although, as stated “This work did not include any data which mandated deposition in public databases.”, it is now common (and soon mandated by NIH) to share datasets such as calcium imaging traces. I do not understand what “Source data are provided with this paper.” Means. Also, add a table of numerical values of results and statistical analyses to supplementary material.

Give number of animals and assays for all N’s

I prefer the terms “animal” “nematode” and “C. elegans” over the term “worm”, as in e.g., the cited Jin, Pokala, Bargmann (2016).

The term “calcium plummet” is unusual and distracting. A more common phrase is “drop in calcium”
The locomotion interneurons AVA and AVB are certainly not “command” interneurons and the study brings evidence that AVB might not even be premotor. I suggest replacing all mentions to “locomotion interneurons”. Moreover, it is important to note (see recommended location on PDF, L71) that there are 2-3 more pairs of locomotion interneurons, namely AVD, PVC, and possibly AVE.

Some citations of reviews of the locomotion circuit are missing (see PDF L180 and other places). Possible references are: Gjorgjieva et al. (2014) *Bioscience*, 64(6), 476-486. doi: 10.1093/biosci/biu058; Zhen & Samuel (2015) *Current opinion in neurobiology*, 33, 117-126 doi: 10.1016/j.conb.2015.03.009; Haspel et al (2020). *Elegantly*. In *The Neural Control of Movement* (pp. 3-29). Academic Press. doi: 10.1016/B978-0-12-816477-8.00001-6

Try and avoid passive voice as much as possible (many places, e.g. L110)

L74-77 Backward locomotion is not equivalent to negative chemotaxis (which is one kind of aversive response that you measure here) and forward not “attractive response”. An increased frequency of reversals translates indirectly to aversion or dwelling.

L75 this study shows that AVB does not “control” B motoneurons.

L174-176 this interpretation is not supported (and not needed here). A speculation whether a common input (proprioceptive or central) or non-linearity of synaptic connection are the source of correlation, could fit here but feels unneeded.

L265 Name the 5 genes that were not tested as mechanoreceptors.

L272 and Fig 4: AIB and RIM traces are not shown because their activity is sync with AVA, but then the sync is lost in the double mutant. To keep Fig 4 simplicity, show these traces in a supplementary figure.

L440,448 (e.g.) for some readers “B- and A-MNs” might be distracting compared to “A- and B-MNs”. (it’s like “pepper and salt” or “dangerous and armed”).

L635 is the Matlab module shared? If not, give details: is bleach correction done by dividing by a fitted exponent?

L646, 650 Consider changing the formulas (and produced plots) so that a decrease gives a negative %

Figures

The differences among the heatmaps in different conditions are not clear. The rows seem to be sorted but it is not clear how. Try sorting the rows by duration, amplitude (max or median), time of first crossing of some value (e.g. 10% of the drop).

Learning-to-speed plots (Fig. 1b and 4e) should either have [His] as x-axis or include horizontal error bars for each data point.

Fig. 1a the text in the diagram is very small (circles can be bigger), the letters ABCD are not useful and can be replaced in the formulas by OP and PA

Fig. 3b add plots for DA and DB.

Fig. 6fj add small diagrams of recording from AVA or AVB (as in 6j) with a text for Ach or Tyr.

Reviewer #3 (Remarks to the Author):

The manuscript by Zhan, Chen et al investigated the role of proprioception in modulating an olfactory learning behavior in *C. elegans*. The authors showed that locomotion speed promotes olfactory learning. They identified stretch receptors acting in the motor neurons and demonstrated that these stretch receptors are required for locomotion to modulate olfactory learning. By electrophysiological experiments, the authors further revealed the property of gap junction between the motor neurons and command interneurons, and identified the gap junction components that function in these electrical synapses. These gap junction components are required for the activation of the command interneurons by stretch stimulation of the motor neurons, and are also essential for olfactory learning.

Overall, the manuscript is well-written, addresses an important question in the field, and provides substantial evidence that supports the roles of locomotion and proprioception in promoting learning.

However, there are several points needed to be further addressed in order to solidify the conclusion and to clarify ambiguity associated with the current manuscript, which I summarize below.

Major comments:

1. The authors stated that inhibition of locomotion by HisCl1 and mutations in the stretch receptors, *asic-1* and *del-1*, reduced PA14 training-induced changes of sensory responses in AVA, AIB, and RIM. However, heatmaps of the Ca²⁺ imaging data (Fig. 1c,d for HisCl1 and Fig. 4c for the stretch receptors) appeared to suggest that the both of these experimental conditions affect the AVA, AIB, and RIM responses of the naive animals rather than those of trained animals. The authors should address whether the changes in the neural responses of the naive animals contribute to the reduction of the amplitude and duration of PA14 training-induced changes. For example, if the amplitude and duration of the neural responses in the naive animals are compared across the conditions (0mM vs 5 mM Histamine, and WT vs *asic-1*; *del-1*), much like the comparison shown in Fig. 1e, f and Fig. 4c, is there a significant difference? If this comparison indicates a difference in the naive condition, then the authors also need to discuss it in light of the observations that inhibition of locomotion and *asic-1*; *del-1* mutations did not alter Naive Choice index.
2. In lines 271-272 and Fig. 4b, the authors showed the data only from AVA for simplicity. The data from AIB and RIM need to be shown.
3. The authors showed that AVB- and AVA-targeted RNAi treatment of the gap junction components affected the learning index (Fig. 5e, f). Given these neurons are thought to be important for the regulation of locomotion by *unc-7* (as indicated by ref. 47), the authors should address whether the effect of learning index is simply due to the reduction in locomotion. The authors should quantify the locomotion of these RNAi-treated animals. Furthermore, if the locomotion is unaffected in these animals, the authors should conduct experiments similar to the one shown in Fig 4e to examine whether RNAi-knockdown of gap junction components reduces or eliminates the modulatory effect of locomotion speed on olfactory learning.
4. Although the authors suggested the role of the disinhibitory circuit among RIM-AVB-AVA in olfactory learning, how the antidromic-rectifying gap junctions between the motor neurons and command interneurons contribute to the neural activity of the interneurons remain unclear. This is largely because the author did not address the effect of AVA-targeted knockdown of *unc-7* and AVB-targeted knockdown of *unc-7* and *inx-19* on the PA14-evoked calcium plummet in AVA, AIB and RIM of naive and trained animals. The authors should add these data to the manuscript and clarify how the gap junction-mediated proprioceptive information affects the interneuronal circuitry for this behavior.
5. As the authors cited (ref. 23), the previous study showed that ASH neurons become strongly activated by PA14 after the training. It is important to address whether inhibition of locomotion, mutations in the stretch receptors, and knockdown of antidromic gap junction components affect the ASH responses in naive and trained animals.

6. The authors demonstrated that the locomotion modulates the PA14 olfactory learning. But whether the effect of the locomotion is specific to this learning paradigm, or the locomotion can broadly affect other types of learning remains unknown. It would be informative to assess whether the locomotion or the stretch receptors regulate other types of learning behaviors in *C. elegans*.

7. Although the current manuscript showed that intervention of locomotion and mutations in the stretch receptors affected olfactory learning, whether the regulation by locomotion occurs in the context of the wild-type animals remains unknown. It would be ideal if the authors could explore and identify a physiological condition under which the wild-type animals alter the locomotion and control the olfactory learning.

Minor comments:

1. line 152: should be "OP50".

2. line 539: should be "RNAi".

Response to Reviewers' Comments

We express our gratitude to the reviewers for their meticulous examination of our manuscript and for providing valuable feedback. Their thoughtful and constructive critiques have been instrumental in improving the quality of our work. We have taken their comments into consideration, conducted additional experiments, and made significant revisions (shown in blue) to the manuscript to address their concerns thoroughly. To facilitate the process, we have provided below our point-by-point responses to the reviewers' comments, with reference to the relevant text line numbers, figure numbers, and reference numbers in the revised manuscript.

REVIEWER COMMENTS

Reviewer #1 (Remarks to the Author):

COMMENT

In Zhan et al. manuscript, the authors set out to address an interesting question on brain functions: how locomotion modulates learning via proprioception. They addressed this question by examining the modulation of three pairs of interneurons AVA, AIB and RIM by locomotion in *C. elegans*. These 3 pairs of neurons are connected with chemical synapses and gap junctions and play important roles in olfactory chemotaxis and its plasticity. Learning-generated changes in the activity and coupling of AVA, AIB and RIM are critical for learned olfactory chemotaxis. They first showed that inhibiting body wall muscles disabled learning and decreased the activities and coupling of AVA, AIB and RIM. To understand how locomotion regulates learning and the activities of AVA, AIB and RIM, they found that ASIC-1 and DEL-1 regulated the response of VB6 and VA5 motor neurons to pressure/mechanical stimulation, naïve and trained activity of AVA, and the enhancement of learning by locomotion. They further showed that VB6 and VA5 motor neurons transmitted information of pressure to AVB and AVA interneurons via gap junctions that were also required for learning. AVB regulate AVA via *acc-2* and respond to tyramine signals via *Igc-55*; these regulations are required for learning. The main contributions of the study include demonstrating the dependence of olfactory learning on locomotion (represented by BWM activity, and mechanical responses in VA and VB motor neurons), mapping the circuit that transmits the locomotory information to interneurons that integrate sensory-motor information to regulate learning, and characterizing the properties of VA and VB motor neurons, and ASIC-1 and DEL-1 expressed in these neurons, as proprioceptive receptors. These findings are new. As a whole, the experimental design is suitable for the question under investigation and the results are compelling.

RESPONSE

We thank the reviewer for this overall assessment of our work.

COMMENT

My main question is on the characterization of VB6 and VA5 motor neurons as proprioceptive receptors. This is the key contribution of the study. They characterized these neurons using “The mechanostimulation was achieved by pressure-ejecting (10 psi, 300 ms)...”. The idea is to use this stimulation pattern to mimic the pressure and stretches generated by the locomotion of the worm. Are the choice of the pressure and duration physiologically reasonable/relevant? This needs to be justified.

RESPONSE

We appreciate the reviewer's thoughtful comment. We selected an injection pressure of 10 psi as it is the minimum pressure that can produce the maximal mechanoreceptor currents (MRCs)

(Supplementary Fig. 5a). This pressure, which corresponds to approximately 220 nN/m² when applied through a glass pipette with a 2 μm tip diameter, falls within the estimated range of internal bending forces experienced by *C. elegans* during locomotion. Specifically, internal bending forces are estimated to range from about 3 nN/m² during swimming to 90 nN/m² while crawling on smooth surfaces, and up to 200-1,000 nN/m² while crawling through an array of flexible pillars (Ref. 37). We chose an injection duration of 300 ms as it allowed us to measure both the steady-state and peak MRCs. To address the reviewer's comments, we have added the following sentences to the manuscript: "We selected an injection pressure of 10 psi (equivalent to ~220 nN/m²) as it was the minimum pressure causing the maximal mechanoreceptor currents (MRCs) (Supplementary Fig. 5a), and falls within the range of estimated internal bending forces experienced by *C. elegans* during locomotion³⁷. We chose an injection duration of 300 ms as it allowed us to measure both the steady-state and peak MRCs." (lines 196-201)

COMMENT

Are these neurons responsive to a range of pressures or durations? A more in-depth characterization of this question would strength the overall contribution of this study.

RESPONSE

As an example, we have tested the responses of VB6 to mechanical stimuli of different pressures (2-14 psi) and durations (30-1,000 ms). We found that increasing the stimulation pressure elevated the peak MRC, which plateaued at 10 psi, while prolonging the stimulation duration increased MRC duration without altering the peak MRC (see Supplementary Fig. 5a, b).

COMMENT

A second question is about the speed-dependency of learning. It is quite intriguing to see an almost linear effect. Does this imply that locomotion of a lower/higher speed (as a result of more/less BWM inhibition) generates weaker/stronger response in VB6 and VA5 motor neurons and subsequently different levels of regulation of AVB and AVA? Does a lower speed in locomotion indicate weaker pressures delivered to the VB6 and VA5 motor neurons during movement or a lower frequency of the mechanical stimulation generated by the movement? If feasible, some experimental tests would be certainly great. At least some discussion and clarifications will be helpful.

RESPONSE

To our knowledge, no studies have yet explored how locomotion speed can affect mechanoreceptor activation or proprioception in *C. elegans*. Therefore, we sought to investigate whether faster locomotion speeds could enhance proprioception by increasing the bending amplitude and/or frequency of *C. elegans*. To this end, we used our automated *C. elegans* tracker (Ref. 34) to measure the body amplitude, a proxy for bending amplitude, and the bending frequency of the animals. Our results showed that the bending frequency increased with higher locomotion speed, while the body amplitude remained constant (Supplementary Fig. 10). These findings suggest that locomotion may improve proprioception by elevating the frequency of mechanoreceptor activation rather than the magnitude, at least in our experimental conditions. In our revised manuscript, we have included a description of these results (lines 320-332).

Minor points:

COMMENT

1. Fig3e,f, it seems that each mutant was used in two types of comparisons – these comparisons and corrections need to be clearly labeled.

RESPONSE

In Fig. 3e and f, we compared all mutant and rescue strains with wild type, and we also compared the rescue strains with their corresponding mutant strains. We performed all statistical comparisons using one-way ANOVA followed by Tukey's post hoc test. We have clarified this information by revising the figure legend.

COMMENT

2. The authors have tested and characterized VB6 and VA5 motor neurons as sensors for mechanical stimuli. Does this lead to the conclusion that other VA and VB neurons are also stretch sensors? The authors may be right, but I suggest using some caution in this generalization.

RESPONSE

The expression of ASIC-1 and DEL-1 in all VA and VB neurons (Fig. 3g) suggests that all the VA and VB motor neurons likely respond to mechanical stimuli. However, because we have tested only one VA and one VB, we agree with the reviewer that it is better to be cautious in the generalization. To address this thoughtful comment, we have added these sentences to the Discussion section: “The expression of ASIC-1 and DEL-1 in all VA and VB suggests that these channels serve similar functions in all these neurons. However, we cannot disregard the possibility that their physiological roles may differ among neurons of the same type since we have only examined one representative VA and one representative VB. To address this possibility, extensive electrophysiological analyses on multiple motor neurons would be required, which is impractical due to the associated time and cost constraints.” (lines 550-556)

3. Fig4e, was any statistical test performed?

RESPONSE

A statistically significant difference existed between the two groups in Figure 4e. However, we inadvertently left out this information in the original manuscript. This error has been corrected in the revised manuscript. We thank the reviewer for raising the question.

Reviewer #2 (Remarks to the Author):

COMMENT

In their manuscript “Locomotion modulates olfactory learning through proprioception in *C. elegans*” the authors follow an intriguing finding that proprioception is involved in aversive learning, to discover the underlying circuit and molecular mechanisms. On that path, they also discover unknown details about the locomotion circuit. The authors use an impressive range of well-chosen methods, from genetics and transgenics, through optogenetic recording and simulation, chemogenetic hyperpolarization, to electrophysiology and behavior analysis. Their approach is thoughtful, though, and methodical. The study is timely, and it provides both specific and generalizable insights that should be of interest to researchers of several fields.

RESPONSE

We thank the reviewer for the overall assessment of our work.

Major comments

COMMENT

Across different sections the authors interchange exercise and locomotion. It is not clear whether they suggest that generic coincidental exercise improves learning capabilities (suggested

by words like “beneficial” and the human-related introduction), or that proprioceptive input is integrated with exteroceptive input to give embodiment, context, and emphasis to the memory. I find the latter much more exciting and worthwhile. The duration and strength of all proprioceptive inputs might serve, for example, as a proxy for overall responsiveness. The vertebrate cerebellum and other examples (given in the discussion) could indeed have similar roles in motor learning; their circuit motif similarities only add to this hypothesis. I suggest rewriting to reduce the “generic coincidental exercise” parts and clarify the “proprioceptive-exteroceptive integration” parts.

RESPONSE

We apologize for using “locomotion” and “exercise” interchangeably in our previous manuscript, as their meanings are not identical. While “locomotion” refers to any movement carried out by muscles, “exercise” generally describes planned and intentional physical activities aimed at achieving beneficial effects. Since it is unclear how proprioception-dependent olfactory learning is related to the animal's “mind”, the term “locomotion” appears to be more appropriate. Therefore, in the revised manuscript, we have replaced “exercise” with “locomotion” wherever it involves *C. elegans*.

We agree with the reviewer that the integration of proprioceptive and exteroceptive inputs to shape memory and other behaviors is an exciting topic. Our results suggest that the AVA, AIB, and RIM interneurons may act as hub neurons to integrate proprioceptive information from motor neurons and exteroceptive information from olfactory sensory neurons to shape olfactory learning behavior. To address the reviewer's insightful comment, the revised manuscript includes the following sentences: "our results indicate that the PA14 training-induced Ca^{2+} drop in AVA, AIB, and RIM interneurons also depends on proprioceptive information from downstream motor neurons, suggesting that interneurons may act as hub neurons to integrate exteroceptive information from olfactory sensory neurons and proprioceptive information from motor neurons to shape olfactory learning."(lines 572-576).

COMMENT

In their narrative and conclusion (including the model in fig. 7) the authors leave out parts of the circuit and focus, without clear justification, on other parts. For example, they describe AVB as the main conduit of ascending input while similar proprioceptive input is mediated by AVA that has similar rectifying GJ with VA. Similarly, RIM directly connects to AVA in addition to the disinhibition via AVB. Finally, proprioceptive MRC in dorsal MN exist even if the mechanoreceptor was not identified; input from DB is demonstrated to be mediated through unc-7 and *inx-19*, but not for DA and not whether they are paired with *unc-9*. Each of these parts should either be included in the narrative and conclusion, or its exclusion should be justified (or justified more clearly).

RESPONSE

We thank the reviewer for these thoughtful comments. For clarity, we address them below in separate paragraphs based on the neurons involved.

1) AVA and AVB

In one of our previous studies (Ref. 27), we described the biophysical properties and molecular compositions of the gap junctions between AVA and A-MNs. However, we acknowledge that we failed to describe the roles of AVA/A-MN and AVB/B-MN gap junctions in a balanced way due to our excitement over new findings. To address the reviewer's comment, we have explicitly described the functions of these two circuits in the revised manuscript: "These results indicate that stretch stimulation of A- and B-MNs is sufficient to activate AVA and AVB through the antidromic-rectifying gap junctions." (lines 387-389). Furthermore, we conducted experiments

using a newly generated triple knockdown strain (*unc-7* in AVA, *unc-7* and *inx-19* in AVB) to assess the combined effects of AVA/A-MN and AVB/B-MN couplings on the PA14-evoked Ca²⁺ changes in AVA, AIB, and RIM, as well as the olfactory learning of PA14 (see Fig. 6 and lines 394-405). The results from these experiments provide further support for the role of antidromic junctional currents in shaping olfactory learning behavior.

2) RIM and AVA

Dual-neuron voltage clamp recordings for RIM and AVA proved to be technically challenging for us. In this study, we did not analyze the interactions between them as we did not anticipate obtaining significant new information by replicating the calcium imaging experiments of previous studies (Ref. 22). To address the reviewer's comment, we added the following paragraph to the Discussion section: "In the *C. elegans* wiring diagram, AVA, AVB, and RIM have many synaptic connections with each other and other neurons^{22,30,50}. Notably, RIM have gap junctions and chemical synapses with AIB and AVA, which also have chemical synapses between them (Fig. 8). Although all these interneurons are involved in olfactory learning, it is not completely understood how they interact at the circuit and synaptic levels. Results from a recent study suggest that INX-4, an innexin expressed in RIM, regulates the olfactory learning of PA14 by forming gap junctions²². Because INX-4 is only expressed in RIM and not in the gap junction partners AIB and AVA^{48,49}, the putative gap junctions are likely heterotypic gap junctions. However, the biophysical properties and molecular compositions of these gap junctions are yet to be determined. Furthermore, the molecular identities and functional properties of the postsynaptic receptors that mediate the putative chemical synaptic transmission among RIM, AIB, and AVA are unclear. Therefore, it is challenging to predict how the RIM-AVB-AVA disinhibitory circuit might interact with the other interneurons in shaping olfactory learning." (lines 585-599).

3) DA and DB motor neurons

We concur with the reviewer's comment that DA and DB motor neurons may also play a role in olfactory learning behavior. However, we did not assess their roles because the mechanoreceptors in these neurons are yet to be identified. Identifying the mechanoreceptors in VA and VB motor neurons has been a lengthy process. To address the reviewer's comment, we added the following sentence to the manuscript: "The representatives of DA and DB, DA4 and DB4, also exhibited mechanosensitivity, although the specific mechanoreceptors remain to be identified. Similar to VA and VB, DA and DB connect to AVA and AVB, respectively¹⁹. This study, along with our previous study²⁷, indicates that both the AVA/A-MN and AVB/B-MN gap junctions act as strong antidromic rectifiers. While they differ in molecular composition, they maintain identical molecular compositions between a specific pair of locomotion interneurons (AVA or AVB) and their connected motor neurons, regardless of whether the motor neurons innervate ventral or dorsal muscles. As VA and VB proprioception regulates olfactory learning behavior by influencing AVA and AVB activities, it is plausible that DA and DB may also contribute, although their precise functions may differ due to their much smaller sustained MRCs compared to VA and VB and their reliance on other mechanosensory receptors." (lines 557-568).

COMMENT

Related issue that was "left behind" is that it seems that *inx-19* has another partner in AVB that is not *unc-9* (because *unc-9(ko)* recapitulate *unc-7(ko)*, not the double mutant L329). It could be *nsy-5* (see Altun et al 2009). If *nsy-5(ko)* exists, the experiments will complete the knowledge. If ko not accessible, address it like the 5 untested mechanoreceptors, by pointing it out and writing that it is not available.

RESPONSE

We would like to apologize for not clarifying the relationship between INX-19 and NSY-5 in the original manuscript. As we have made clear in the revised version (line 357), NSY-5 is actually another name for INX-19. In the original manuscript, we described that UNC-7 and INX-19 act in AVB while UNC-9 acts in B-MNs to mediating the electrical coupling between AVB and B-MNs. However, we acknowledge the reviewer's comment that there must be another innexin(s) involved in B-MNs, as the AVB-VB6 coupling was deficient but not completely absent in the *unc-9* mutant. To address this issue, we investigated the potential involvement of six other innexins known to be expressed in B-MNs, including *inx-1*, *inx-7*, *inx-10*, *inx-14*, *inx-3*, and *inx-12*, by comparing junctional currents between wild-type and mutant strains (for *inx-1*, *inx-7*, *inx-10*, and *inx-14*) or knockdown strains (for *inx-3* and *inx-12*). However, our results showed that none of the mutants or knockdown strains displayed a deficiency in electrical coupling compared to wild type, suggesting the involvement of an innexin that has yet to be identified in B-MNs. These new findings are described in the Results section (lines 367-376) and presented in Supplementary Fig. 11c, d.

COMMENT

How did the author restrict the mechanical stimulation to those neurons? For example, when stimulating either VB6 or VA5 in Fig 5, how did they aim for one or the other. Similarly, when recording from a stimulated MN in Fig. 3, how did the author exclude mechanical stimulation of neighboring MN that have synaptic connections to the recorded MN (for example, the soma of VB6 overlaps in the ventral nerve cord with the NMJ area of VB5 and the asynaptic neurite that extends after the axons of VB1-4)?

RESPONSE

As shown in Fig. 3a and described in the Results section (lines 201-203), we stimulated motor neurons mechanically by puffing the bath solution through a glass pipette, which was aimed at the ventral nerve cord perpendicularly so that the soma, axon, or dendrite of a specific motor neuron could be stimulated specifically. However, due to the proximity of neighboring motor neuron somata and the fact that motor neuron axons and dendrites run together in the ventral nerve cord, it was impossible to restrict the mechanical stimulation to a single motor neuron. Nevertheless, the facts that little MRC was evoked by axonal stimulation and the peak MRC caused by dendritic stimulation diminished with increasing distances from the soma suggest that the MRC mainly resulted from mechanical stimulation of the recorded neuron. In the revised manuscript, we acknowledge the limitation of our technique by including the following sentences: "Due to the proximity of neighboring motor neuron somata and the fact that motor neuron axons and dendrites run together in the ventral nerve cord, it was impossible to restrict the mechanical stimulation to a single motor neuron. However, the fact that MRCs could be induced only by stimulating specific regions of the motor neuron indicates that the recorded MRCs resulted from mechanoreceptors of the same neuron." (lines 220-225).

COMMENT

The dorsal MN produced MRC without a sustained current (Fig. 3) compared to the ventral MNs. Did the author stimulate the commissure and dorsal cord portions of these MNs?

RESPONSE

We could not stimulate the commissures or dorsal portions of these MNs because they were either severed by the diamond dissecting tool or damaged by the glue used to immobilize the animal for electrophysiological recordings. To address the reviewer's thoughtful comment, we mechanically stimulated the dendrites of DA4 and DB4 at various distances from the soma, which could be done because DA4 and VA4 dendrites are located on the ventral side (see Supplementary Fig. 6a). We observed reduced peak MRC with increasing distance from the soma

(see Supplementary Fig. 6b, c), which resembled the dendritic responses of VA5 and VB6 (see Fig. 3a, b). In our revised manuscript, we have included a description of these results (lines 207-213).

COMMENT

The causality of AVA-AVB anticorrelation (e.g. L405) and learning is not supported. They could each be caused by proprioception that was demonstrated necessary for both.

RESPONSE

The AVA-targeted *acc-2* knockdown caused significant decreases in both the AVA-AVB anticorrelation coefficient and the olfactory learning index (Fig. 7h, i), suggesting that ACC-2-dependent inhibition of AVA by AVB is important to both AVA-AVB anticorrelation and olfactory learning. To address the reviewer's comment, we changed our conclusion to "Thus, ACC-2-dependent inhibition of AVA by AVB is necessary for their anticorrelation and for olfactory learning." (lines 470-471).

Minor comments:

(More comments and edits are also on attached PDF file)

RESPONSE

We would like to express our gratitude to the reviewer for the meticulous editing of our manuscript. We have diligently incorporated the reviewer's edits throughout the revised manuscript. However, concerning the comment about the cited references in line 79, we were unable to locate any publication by R Durbin on motor neuron stretch receptors. In the White et al (1986) paper, the authors referenced personal communications with L. Byerly and R. L. Russell as a source of speculation on motor neurons as stretch sensors. We have revised the manuscript accordingly to address this comment from the reviewer. The relevant sentences now state, "The *C. elegans* motor circuit lacks specialized proprioceptive sensory neurons. L. Byerly and R.L. Russell speculated many years ago that A- and B-MNs might function as stretch sensors, utilizing their long asynaptic neurites³⁰." (lines 76-78). We would cite Richard Durbin's publication if the reviewer would kindly take the trouble to help us identify it.

COMMENT

L656-657 Although, as stated "This work did not include any data which mandated deposition in public databases.", it is now common (and soon mandated by NIH) to share datasets such as calcium imaging traces. I do not understand what "Source data are provided with this paper." Means. Also, add a table of numerical values of results and statistical analyses to supplementary material.

Give number of animals and assays for all N's

RESPONSE

We addressed the reviewer's comment by taking the following actions: Firstly, we have provided all data generated from this study in the Excel format as "Source data" files, with each file corresponding to one figure. Secondly, we have modified the "Data Availability" statement to state that "All data generated or analyzed during this study are included in this published article and its supplementary information files. Raw data are provided as source data files in reference to the main and supplementary figures." (lines 776-779). Thirdly, we have described the statistical methods and p values in figure legends, and provided numerical values for results and statistical analyses to the source data files. Finally, we have included the sample size (*n*) for all groups in each figure.

COMMENT

I prefer the terms “animal” “nematode” and “C. elegans” over the term “worm”, as in e.g., the cited Jin, Pokala, Bargmann (2016).

The term “calcium plummet” is unusual and distracting. A more common phrase is “drop in calcium”

The locomotion interneurons AVA and AVB are certainly not “command” interneurons and the study brings evidence that AVB might not even be premotor. I suggest replacing all mentions to “locomotion interneurons”. Moreover, it is important to note (see recommended location on PDF, L71) that there are 2-3 more pairs of locomotion interneurons, namely AVD, PVC, and possibly AVE.

Some citations of reviews of the locomotion circuit are missing (see PDF L180 and other places).

Possible references are: Gjorgjieva et al. (2014) *Bioscience*, 64(6), 476-486. doi:

10.1093/biosci/biu058; Zhen & Samuel (2015) *Current opinion in neurobiology*, 33, 117-126 doi:

10.1016/j.conb.2015.03.009; Haspel et al (2020). *Elegantly*. In *The Neural Control of Movement* (pp. 3-29). Academic Press. doi: 10.1016/B978-0-12-816477-8.00001-6

Try and avoid passive voice as much as possible (many places, e.g. L110)

RESPONSE

We thank the reviewer for these suggestions and comments. In response, we have made the following changes throughout the manuscript: Firstly, we have replaced the term “worm” with “animal”, “calcium plummet” with “calcium drop”, and “command interneurons” with “locomotion interneurons”. Secondly, we have added AVD, AVE, and PVC to our description of the locomotion interneurons in the Introduction. Thirdly, we have provided the missing references. Finally, we have changed the passive voice to the active voice wherever appropriate.

COMMENT

L74-77 Backward locomotion is not equivalent to negative chemotaxis (which is one kind of aversive response that you measure here) and forward not “attractive response”. An increased frequency of reversals translates indirectly to aversion or dwelling.

L75 this study shows that AVB does not “control” B motoneurons.

RESPONSE

We agree with the reviewer's comment and have made the necessary revision to the sentence, which now reads: “AVA initiate backward locomotion by controlling A-type cholinergic motor neurons (A-MNs) through both chemical synapses and gap junctions²⁷, whereas AVB promote forward locomotion only through gap junctions with B-type cholinergic motor neurons (B-MNs)¹⁹.” (line 72-75)

L174-176 this interpretation is not supported (and not needed here). A speculation whether a common input (proprioceptive or central) or non-linearity of synaptic connection are the source of correlation, could fit here but feels unneeded.

RESPONSE

We have deleted this sentence.

L265 Name the 5 genes that were not tested as mechanoreceptors.

RESPONSE

In this study, we identified ASIC-1 and DEL-1 as the mechanoreceptors in VA5 and VB6 by screening available mutants for 25 DEG/ENaC/ASIC genes. However, for DA4 and DB4, we

only tested the *asic-1;del-1* double mutant and did not observe any significant effect of the mutations on MRCs. We have clarified this point in the revised manuscript (lines 275-276).

COMMENT

L272 and Fig 4: AIB and RIM traces are not shown because their activity is sync with AVA, but then the sync is lost in the double mutant. To keep Fig 4 simplicity, show these traces in a supplementary figure.

RESPONSE

As suggested, a new figure (Supplementary Fig. 8) shows the data from AIB and RIM.

COMMENT

L440,448 (e.g.) for some readers “B- and A-MNs” might be distracting compared to “A- and B-MNs”. (it’s like “pepper and salt” or “dangerous and armed”).

RESPONSE

We have changed “B- and A-MNs” to “A- and B-MNs” throughout the manuscript.

COMMENT

L635 is the Matlab module shared? If not, give details: is bleach correction done by dividing by a fitted exponent?

RESPONSE

The MATLAB module employed in this study utilizes a cubic interpolation algorithm to generate a curve along the user-selected baseline. The raw data is then divided by the interpolated points of the baseline to produce the normalized data. To address the reviewer's comment, we added this clause to an existing sentence in the Methods section: “...., which generates the corrected baseline and F/F₀ peaks using a cubic interpolation algorithm (see Code Availability statement for further information).” (lines 754-756). We have uploaded the software to our lab's website and provided step-by-step instructions on how to use the Matlab module. We now state in Code availability that “The MATLAB module used to correct photobleaching-induced drop of fluorescence signal, along with user instructions, is freely available at <https://health.uconn.edu/worm-lab/track-a-worm/>.” (lines 781-784).

COMMENT

L646, 650 Consider changing the formulas (and produced plots) so that a decrease gives a negative %

RESPONSE

As the reviewer suggested, we have changed the formulas to $\left(\frac{\text{amplitude of trained animals}}{\text{averaged amplitude of naïve animals}} - 1\right) \times 100\%$ and $\left(\frac{\text{duration of trained animals}}{\text{averaged duration of naïve animals}} - 1\right) \times 100\%$, respectively. The related plots have been revised accordingly.

COMMENT

Figures

The differences among the heatmaps in different conditions are not clear. The rows seem to be sorted but it is not clear how. Try sorting the rows by duration, amplitude (max or median), time of first crossing of some value (e.g. 10% of the drop).

RESPONSE

We have sorted the rows of heat maps based on the response duration in all the related figures.

COMMENT

Learning-to-speed plots (Fig. 1b and 4e) should either have [His] as x-axis or include horizontal error bars for each data point.

RESPONSE

As the reviewer suggested, we have added horizontal error bars for each data point in the revised Fig. 1b and Fig. 4e.

COMMENT

Fig. 1a the text in the diagram is very small (circles can be bigger), the letters ABCD are not useful and can be replaced in the formulas by OP and PA

RESPONSE

We have modified Fig. 1a as suggested.

COMMENT

Fig. 3b add plots for DA and DB.

RESPONSE

In the original manuscript, we only recorded MRCs of DA4 and DB4 in response to soma stimulation (Fig. 3c). In response to the reviewer's comment, we conducted new experiments to stimulate the dendrites of DA4 and DB4 at various distances from the soma. Our findings showed a decrease in peak MRC with increasing distance from the soma (Supplementary Fig. 6b, c), which was similar to the results from VA5 and VB6 (Fig. 3a, b). Unfortunately, we were unable to stimulate the axons of these two neurons because they project to the dorsal side as circumferential commissures, which were either severed by the diamond dissecting tool or damaged by the glue used to immobilize the animals.

COMMENT

Fig. 6fj add small diagrams of recording from AVA or AVB (as in 6j) with a text for Ach or Tyr.

RESPONSE

We have revised Fig. 6f, j (now Fig. 7f, j) and Supplementary Fig. 12 as suggested.

Reviewer #3 (Remarks to the Author):

COMMENT

The manuscript by Zhan, Chen et al investigated the role of proprioception in modulating an olfactory learning behavior in *C. elegans*. The authors showed that locomotion speed promotes olfactory learning. They identified stretch receptors acting in the motor neurons and demonstrated that these stretch receptors are required for locomotion to modulate olfactory learning. By electrophysiological experiments, the authors further revealed the property of gap junction between the motor neurons and command interneurons, and identified the gap junction components that function in these electrical synapses. These gap junction components are required for the activation of the command interneurons by stretch stimulation of the motor neurons, and are also essential for olfactory learning.

Overall, the manuscript is well-written, addresses an important question in the field, and provides

substantial evidence that supports the roles of locomotion and proprioception in promoting learning. However, there are several points needed to be further addressed in order to solidify the conclusion and to clarify ambiguity associated with the current manuscript, which I summarize below.

RESPONSE

We thank the reviewer for the overall assessment of our work.

Major comments:

COMMENT

1. The authors stated that inhibition of locomotion by HisCl1 and mutations in the stretch receptors, *asic-1* and *del-1*, reduced PA14 training-induced changes of sensory responses in AVA, AIB, and RIM. However, heatmaps of the Ca²⁺ imaging data (Fig. 1c,d for HisCl1 and Fig. 4c for the stretch receptors) appeared to suggest that the both of these experimental conditions affect the AVA, AIB, and RIM responses of the naïve animals rather than those of trained animals. The authors should address whether the changes in the neural responses of the naïve animals contribute to the reduction of the amplitude and duration of PA14 training-induced changes. For example, if the amplitude and duration of the neural responses in the naïve animals are compared across the conditions (0mM vs 5 mM Histamine, and WT vs *asic-1*; *del-1*), much like the comparison shown in Fig. 1e, f and Fig. 4c, is there a significant difference? If this comparison indicates a difference in the naïve condition, then the authors also need to discuss it in light of the observations that inhibition of locomotion and *asic-1*; *del-1* mutations did not alter Naïve Choice index.

RESPONSE

As the reviewer suggested, we determined the effects of histamine application and the *asic-1*; *del-1* double mutations on PA14-induced Ca²⁺ drop in the AVA, AIB, and RIM interneurons of naïve animals. Both histamine application and the *asic-1*; *del-1* double mutations attenuated the PA14-induced Ca²⁺ drop significantly in all the interneurons. The double mutations also had the additional effect of significantly shortening the duration of the PA14-induced Ca²⁺ drop (Supplementary Fig. 4). We added the following sentence to the manuscript: “Further analyses revealed that histamine minimized the difference in the Ca²⁺ drop between naïve and trained animals by mainly acting on naïve animals (Supplementary Fig. 4a).” (lines 165-167). In the Discussion section, we discussed why inhibition of locomotion by histamine and *asic-1*; *del-1* mutations did not alter the naïve choice index: “Our results indicate that motor neuron-mediated proprioception does not influence the naïve choice index. The preference of *C. elegans* for PA14 in the naïve state is thought to be determined by the intrinsic properties of AWB and AWC sensory neurons²¹. However, these sensory neurons do not receive synaptic inputs from either A- and B-MNs or the AVA, AIB, RIM, and AVB interneurons³⁰, which may explain why proprioception did not affect the naïve choice index.” (lines 600-605)

COMMENT

2. In lines 271-272 and Fig. 4b, the authors showed the data only from AVA for simplicity. The data from AIB and RIM need to be shown.

RESPONSE

As suggested, a new figure (Supplementary Fig. 8) shows the data from AIB and RIM.

COMMENT

3. The authors showed that AVB- and AVA-targeted RNAi treatment of the gap junction components affected the learning index (Fig. 5e, f). Given these neurons are thought to be important for the regulation of locomotion by *unc-7* (as indicated by ref. 47), the authors should address whether the effect of learning index is simply due to the reduction in locomotion. The authors should quantify the locomotion of these RNAi-treated animals. Furthermore, if the locomotion is unaffected in these animals, the authors should conduct experiments similar to the one shown in Fig 4e to examine whether RNAi-knockdown of gap junction components reduces or eliminates the modulatory effect of locomotion speed on olfactory learning.

RESPONSE

To address this comment, we compared locomotion speed among different strains: wild type, AVB-targeted *unc-7* and *inx-19* knockdown strain, AVA-targeted *unc-7* knockdown strain, and a newly created triple knockdown strain targeting *unc-7* and *inx-19* in AVB and *unc-7* in AVA. The AVA-targeted *unc-7* knockdown strain showed similar locomotion speed to wild type (142.63 $\mu\text{m/s}$ vs. 151.74 $\mu\text{m/s}$, $p = 0.529$). However, both the AVB-targeted *unc-7* knockdown strain (135.61 $\mu\text{m/s}$) and the triple knockdown strain (133.24 $\mu\text{m/s}$) exhibited significantly slower locomotion speed compared to wild type. The magnitude of locomotion reduction caused by AVB-targeted *unc-7* and *inx-19* knockdown was similar to that caused by 0.02 mM histamine in the BWM-HisC11 strain, which had no significant effect on the olfactory learning index (Fig. 1b). Therefore, the olfactory learning defect observed in the knockdown strains resulted from blockade of the antidromic junctional current rather than inhibition of locomotion. We have added similar descriptions to the manuscript (lines 406-421).

COMMENT

4. Although the authors suggested the role of the disinhibitory circuit among RIM-AVB-AVA in olfactory learning, how the antidromic-rectifying gap junctions between the motor neurons and command interneurons contribute to the neural activity of the interneurons remain unclear. This is largely because the author did not address the effect of AVA-targeted knockdown of *unc-7* and AVB-targeted knockdown of *unc-7* and *inx-19* on the PA14-evoked calcium plummet in AVA, AIB and RIM of naive and trained animals. The authors should add these data to the manuscript and clarify how the gap junction-mediated proprioceptive information affects the interneuronal circuitry for this behavior.

RESPONSE

To address the reviewer's insightful comment, we generated a triple knockdown strain targeting *unc-7* in AVA and both *unc-7* and *inx-19* in AVB. Our results showed that the training-induced attenuation of Ca^{2+} drop in AVA, AIB, and RIM was significantly smaller in amplitude and shorter in duration in the triple knockdown strain compared to wild type. Additionally, the triple RNAi strain showed approximately 50% decrease in the learning index without a change in the naïve choice index, and a greatly reduced effect of locomotion speed on the learning index compared with wild type. These findings support our conclusion that antidromic-rectifying gap junctions play a critical role in olfactory learning. The new results are described in the Results section (lines 394-405) and shown in Fig. 6.

COMMENT

5. As the authors cited (ref. 23), the previous study showed that ASH neurons become strongly activated by PA14 after the training. It is important to address whether inhibition of locomotion, mutations in the stretch receptors, and knockdown of antidromic gap junction components affect the ASH responses in naive and trained animals.

RESPONSE

To address the reviewer's suggestion, we compared PA14-evoked responses in ASH among several groups, including wild type, the BWM-HisC11 strain treated with histamine (5 mM), the *asic-1;del-1* double mutant, and the triple knockdown strain (*unc-7* in AVA and *unc-7* and *inx-19* in AVB), in both naïve and trained animals. Consistent with a previous report (Ref. 22), we observed much stronger Ca²⁺ signals in trained animals compared to naïve ones. However, our analysis revealed that ASH responses were similar across all the groups, whether naïve or trained. These results have been described in the Results section (lines 494-507) and are presented in Supplementary Fig. 13.

COMMENT

6. The authors demonstrated that the locomotion modulates the PA14 olfactory learning. But whether the effect of the locomotion is specific to this learning paradigm, or the locomotion can broadly affect other types of learning remains unknown. It would be informative to assess whether the locomotion or the stretch receptors regulate other types of learning behaviors in *C. elegans*.

RESPONSE

To address the reviewer's insightful comment, we investigated the impact of *asic-1* and *del-1* mutations on olfactory adaptation. Our findings revealed that either *asic-1* or *del-1* mutation significantly impaired olfactory adaptation to benzaldehyde, with the effect of double mutations being more severe than either single mutation ($p < 0.01$). The defect in the *asic-1;del-1* double mutant was partially rescued by co-expressing wild-type *asic-1* and *del-1* in VA and VB using *Pdel-1*, indicating that ASIC-1 and DEL-1 regulate olfactory adaptation by acting in cholinergic motor neurons as well as other unidentified neurons. These new results are described in the Results section (lines 509-528) and shown in Supplementary Fig. 14.

COMMENT

7. Although the current manuscript showed that intervention of locomotion and mutations in the stretch receptors affected olfactory learning, whether the regulation by locomotion occurs in the context of the wild-type animals remains unknown. It would be ideal if the authors could explore and identify a physiological condition under which the wild-type animals alter the locomotion and control the olfactory learning.

RESPONSE

We are grateful to the reviewer for this suggestion. To address it, we used gelatin at various concentrations on standard nematode growth medium plates to modulate the locomotion speed of wild-type animals to different extents (Ref. 35). Our results revealed that higher locomotion speed increased the olfactory learning index without affecting the naïve choice index, as shown in Supplementary Fig. 2. This finding further corroborates the conclusion we reached with the HisC11 transgenic strain. These new results are described in the Results section (lines 142-148).

Minor comments:

COMMENT

1. line 152: should be "OP50".

RESPONSE

The error has been corrected.

COMMENT

2. line 539: should be "RNAi".

RESPONSE

The error has been corrected.

REVIEWER COMMENTS

Reviewer #1 (Remarks to the Author):

In the revised manuscript, the authors addressed my questions and concerns using new experimental results or by including thoughtful discussions. I now support the publication of the paper.

Reviewer #2 (Remarks to the Author):

The authors invested time and attention to improve the manuscript following the feedback. Their enthusiasm to share data and original code in this version is commendable. The manuscript is now more clear and the new results following reviewers' comments make the story more complete.

The authors replaced the term "exercise" in places where it meant locomotion in the nematode. The first paragraph of the introduction begins with the word locomotion but then still leads the reader toward a coincidental generic exercise. I leave it to the authors to decide but I suggest that, if this "mammalian justification" has to be included, it belongs in the discussion.

Citing White et al 1986 to give credit to Byerly and Russell is a good reference (as was cited in <https://doi.org/10.1016/B978-0-12-816477-8.00001-6>). The speculations in R. Durbin's thesis (<https://www.wormatlas.org/Durbin/Durbin1987partII.html>) which I suggested are not as specific.

L201 Another reason for an injection duration of 300 ms is that it is within the range of bend duration (~0.5 - 2 Hz give 250-1,000 ms).

L766 and L769 Use "mean" in both numerator and denominator or in neither (I prefer neither). As written, it is unclear whether the values measured in trained animals were averaged.

There are some minor grammar issues in the old and new text that could be easily found and corrected with a service such as Grammarly.

Gal Haspel

Reviewer #3 (Remarks to the Author):

The authors conducted superb jobs on revising the manuscript. They added new results, and the revised manuscript was substantially improved. The authors addressed quite convincingly almost all the concerns I had on the initial manuscript. However, I have a few points that need to be addressed, all of which are related to my comment #1 on the initial manuscript.

The authors' new analysis of the calcium responses (Supplemental Figure 4) now revealed that the histamine application and the *asic-1; del-1* double mutations affected mainly on the response in the naive animals rather than the trained animals. Similar trends were also observed in their newly added data: triple RNAi strain (Figure 6) and AVB-HisCl1 strain (Figure7). Given these observations, I would like to suggest the following points.

1. Lines 293-298. The authors should call Supplemental Figure 4 and state that the *asic-1; del-1* double mutations affected the response in naive animals.
2. In new Figure 6, lines 394-400. Similar to the analysis in Supplemental Figure 4, the authors should determine the effect of triple RNAi on naive (and trained) animals.
3. In new Figure7, lines 428-431. Again, similar to the analysis in Supplemental Figure 4, the authors should determine the effect of AVE-HisCl1 on naive (and trained) animals.
4. Lines 600-605. I appreciate it that the authors added this new discussion. However, I think what would be more informative and helpful is to discuss how a change in the calcium drop of naive animals could lead to the change in the learned behavior (Learning index) rather than the naive behavior (Naive choice index). Adding some discussion (even short one) would be helpful.

A minor comment: Line 363, *unc-7(e6)* should be *unc-7(e5)*.

REVIEWER COMMENTS

Reviewer #1 (Remarks to the Author):

In the revised manuscript, the authors addressed my questions and concerns using new experimental results or by including thoughtful discussions. I now support the publication of the paper.

Reviewer #2 (Remarks to the Author):

The authors invested time and attention to improve the manuscript following the feedback. Their enthusiasm to share data and original code in this version is commendable. The manuscript is now more clear and the new results following reviewers' comments make the story more complete.

The authors replaced the term “exercise” in places where it meant locomotion in the nematode. The first paragraph of the introduction begins with the word locomotion but then still leads the reader toward a coincidental generic exercise. I leave it to the authors to decide but I suggest that, if this “mammalian justification” has to be included, it belongs in the discussion.

Citing White et al 1986 to give credit to Byerly and Russell is a good reference (as was cited in <https://doi.org/10.1016/B978-0-12-816477-8.00001-6>). The speculations in R. Durbin's thesis (<https://www.wormatlas.org/Durbin/Durbin1987partII.html>) which I suggested are not as specific.

L201 Another reason for an injection duration of 300 ms is that it is within the range of bend duration (~0.5 - 2 Hz give 250-1,000 ms).

L766 and L769 Use “mean” in both numerator and denominator or in neither (I prefer neither). As written, it is unclear whether the values measured in trained animals were averaged.

There are some minor grammar issues in the old and new text that could be easily found and corrected with a service such as Grammarly.

Gal Haspel

Reviewer #3 (Remarks to the Author):

The authors conducted superb jobs on revising the manuscript. They added new results, and the revised manuscript was substantially improved. The authors addressed quite convincingly almost all the concerns I had on the initial manuscript. However, I have a few points that need to be addressed, all of which are related to my comment #1 on the initial manuscript.

The authors' new analysis of the calcium responses (Supplemental Figure 4) now revealed that the histamine application and the *asic-1*; *del-1* double mutations affected mainly on the response in the naive animals rather than the trained animals. Similar trends were also observed in their newly added data: triple RNAi strain (Figure 6) and AVB-HisC11 strain (Figure7). Given these observations, I would like to suggest the following points.

1. Lines 293-298. The authors should call Supplemental Figure 4 and state that the *asic-1*; *del-1* double mutations affected the response in naive animals.
2. In new Figure 6, lines 394-400. Similar to the analysis in Supplemental Figure 4, the authors should determine the effect of triple RNAi on naive (and trained) animals.
3. In new Figure 7, lines 428-431. Again, similar to the analysis in Supplemental Figure 4, the authors should determine the effect of AVE-HisC11 on naive (and trained) animals.
4. Lines 600-605. I appreciate it that the authors added this new discussion. However, I think what would be more informative and helpful is to discuss how a change in the calcium drop of naive animals could lead to the change in the learned behavior (Learning index) rather than the naive behavior (Naive choice index). Adding some discussion (even short one) would be helpful.

A minor comment: Line 363, *unc-7(e6)* should be *unc-7(e5)*.

Response to Reviewers' Comments

We are thrilled to receive feedback from all the reviewers, who have unanimously acknowledged the substantial improvements made to our revised manuscript. We are pleased to note that we have adequately addressed all their comments on the original version. We extend our gratitude to Reviewers 2 and 3 for their valuable additional suggestions on the revised manuscript. We have taken their suggestions and made revisions (shown in blue) to the manuscript. In the following section, we provide a detailed point-by-point response to each of the reviewers' suggestions.

Reviewer #1 (Remarks to the Author):

In the revised manuscript, the authors addressed my questions and concerns using new experimental results or by including thoughtful discussions. I now support the publication of the paper.

RESPONSE

We thank the reviewer for supporting the publication of our paper.

Reviewer #2 (Remarks to the Author):

The authors invested time and attention to improve the manuscript following the feedback. Their enthusiasm to share data and original code in this version is commendable. The manuscript is now more clear and the new results following reviewers' comments make the story more complete.

RESPONSE

We appreciate the reviewer's comment.

The authors replaced the term "exercise" in places where it meant locomotion in the nematode. The first paragraph of the introduction begins with the word locomotion but then still leads the reader toward a coincidental generic exercise. I leave it to the authors to decide but I suggest that, if this "mammalian justification" has to be included, it belongs in the discussion.

RESPONSE

We have replaced the word "exercise" with "locomotion" throughout the first paragraph of the Introduction.

Citing White et al 1986 to give credit to Byerly and Russell is a good reference (as was cited in <https://doi.org/10.1016/B978-0-12-816477-8.00001-6>). The speculations in R. Durbin's thesis (<https://www.wormatlas.org/Durbin/Durbin1987partII.html>) which I suggested are not as specific.

RESPONSE

We appreciate this comment.

L201 Another reason for an injection duration of 300 ms is that it is within the range of bend duration (~0.5 - 2 Hz give 250-1,000 ms).

RESPONSE

We appreciate the reviewer's comment. We previously found that the bending frequency of wild-type animals is approximately 0.4 events per sec during crawling and approximately 4 events per sec during swimming (Ref. 34 and 38). This translates to bending durations of approximately 2,500 ms and 250 ms, respectively. The manuscript now has this revised sentence: "We chose an injection duration of 300 ms as it falls within the range of *C. elegans* bending duration (250 to 2,500 ms based on reported bending frequencies of approximately 0.4 to 4 events per sec)^{34,38}. This duration allowed us to capture both the peak and steady-state MRCs." (lines 200-203).

L766 and L769 Use "mean" in both numerator and denominator or in neither (I prefer neither). As written, it is unclear whether the values measured in trained animals were averaged.

RESPONSE

We have followed the suggestion and removed the word "averaged" from the formulas as recommended.

There are some minor grammar issues in the old and new text that could be easily found and corrected with a service such as Grammarly.

RESPONSE

We have utilized Grammarly to check the grammar of our manuscript and have addressed the minor grammatical issues accordingly. We express our gratitude to the reviewer for providing this valuable suggestion.

Reviewer #3 (Remarks to the Author):

The authors conducted superb jobs on revising the manuscript. They added new results, and the revised manuscript was substantially improved. The authors addressed quite convincingly almost all the concerns I had on the initial manuscript. However, I have a few points that need to be addressed, all of which are related to my comment #1 on the initial manuscript.

RESPONSE

We appreciate the reviewer's comment.

The authors' new analysis of the calcium responses (Supplemental Figure 4) now revealed that the histamine application and the *asic-1; del-1* double mutations affected mainly on the response in the naive animals rather than the trained animals. Similar trends were also observed in their newly added data: triple RNAi strain (Figure 6) and AVB-HisC11 strain (Figure7). Given these observations, I would like to suggest the following points.

1. Lines 293-298. The authors should call Supplemental Figure 4 and state that the *asic-1; del-1* double mutations affected the response in naive animals.
2. In new Figure 6, lines 394-400. Similar to the analysis in Supplemental Figure 4, the authors should determine the effect of triple RNAi on naive (and trained) animals.
3. In new Figure7, lines 428-431. Again, similar to the analysis in Supplemental Figure 4, the authors should determine the effect of AVE-HisC11 on naive (and trained) animals.

RESPONSE

Following the suggestion, we have quantified the differences in the Ca²⁺ drop between the wild type and the triple knockdown strain, as well as between the wild type and the AVB-HisCl1 strain, in both naïve and trained animals. The newly obtained results have been incorporated into Supplementary Fig. 4. The manuscript now includes the following revised or newly added sentences:

1. “The primary effect of the *asic-1;del-1* double mutation was to reduce the Ca²⁺ drop in naïve animals (Supplementary Fig. 4b).” (lines 300-301).
2. “The triple RNAi strain exhibited great reductions in both the magnitude and duration of the PA14 training-induced Ca²⁺ drop in AVA, AIB, and RIM when compared to wild type (Fig. 6a-d), predominantly resulting from a reduced Ca²⁺ drop in naïve animals (Supplementary Fig. 4c).” (lines 402-405).
3. “The inhibition of AVB primarily affected training-induced Ca²⁺ changes by acting on naïve animals (Supplementary Fig. 4d).” (lines 438-439).

4. Lines 600-605. I appreciate it that the authors added this new discussion. However, I think what would be more informative and helpful is to discuss how a change in the calcium drop of naïve animals could lead to the change in the learned behavior (Learning index) rather than the naïve behavior (Naïve choice index). Adding some discussion (even short one) would be helpful.

RESPONSE

To address this comment of the reviewer, we have added the following paragraph to the Discussion: “The PA14-induced Ca²⁺ drop was significantly reduced in wild-type animals after PA14 training. However, this reduction was not observed in animals with impaired motor neuron-mediated proprioception. Our quantitative analyses showed that wild-type animals differed from those with locomotion inhibition or impairment of the motor neuron-mediated proprioception circuit primarily in terms of having a significantly larger PA14-induced Ca²⁺ drop in AVA, AIB, and RIM during the naïve stage, with minimal differences in the post-PA14 training stage. These analyses indicate that the PA14-induced Ca²⁺ drop has both a proprioception-dependent and a proprioception-independent component, and that the occurrence of olfactory learning depends mainly on whether there is a significant Ca²⁺ drop between the naïve and post-PA14 training stages. Consistently, a mutation of *nmr-1*, which encodes the *C. elegans* homolog of the NR1 subunit of mammalian NMDA receptors, also impairs PA14 olfactory learning and affects the PA14-induced Ca²⁺ drop in AVA, AIB, and RIM mainly during the naïve stage²². One question that has arisen from our analyses of the Ca²⁺ drop is why inhibition of the motor neuron-mediated proprioception circuit did not alter the naïve choice index. This is likely because the naïve preference of *C. elegans* for PA14 is determined by the intrinsic properties of AWB and AWC sensory neurons²¹. However, these sensory neurons do not receive synaptic inputs from either A- and B-MNs or the AVA, AIB, RIM, and AVB interneurons³⁰.” (lines 608-626)

A minor comment: Line 363, *unc-7(e6)* should be *unc-7(e5)*.

RESPONSE

We thank the reviewer for bringing this error to our attention. We have rectified it accordingly.

REVIEWERS' COMMENTS

Reviewer #3 (Remarks to the Author):

The authors addressed my questions and concerns. I now support the publication of the manuscript.

REVIEWERS' COMMENTS

Reviewer #3 (Remarks to the Author):

The authors addressed my questions and concerns. I now support the publication of the manuscript.

Response to Reviewers' Comments

Reviewer #3 (Remarks to the Author):

The authors addressed my questions and concerns. I now support the publication of the manuscript.

RESPONSE

We thank the reviewer for supporting the publication of our manuscript.